



# A method to assess the accuracy of sonic anemometer measurements

Alfredo Peña[1], Ebba Dellwik[1], and Jakob Mann[1]

[1]DTU Wind Energy, Technical University of Denmark, Roskilde, Denmark

*Correspondence to:* Alfredo Peña (aldi@dtu.dk)

**Abstract.** We propose a method to assess the accuracy of atmospheric turbulence measurements performed by sonic anemometers and test it by analysis of measurements from two commonly used sonic anemometers, a Metek USA-1 and a Campbell CSAT3, at two locations in Denmark. The method relies on the estimation of the ratio of the vertical to the along-wind velocity power spectrum within the inertial subrange. When we correct the USA-1 to account for three-dimensional flow-distortion effects, as recommended by Metek GmbH, the ratio is very close to 4/3 as expected from Kolmogorov's hypothesis, whereas non-corrected data show a ratio close to 1. For the CSAT3, non-corrected data show a ratio close to 1.1 for the two sites and for wind directions where the instrument is not directly affected by the mast. After applying a previously suggested flow-distortion correction, the ratio increases up to ≈1.2, implying that the effect of flow distortion in this instrument is still not properly accounted for.

## 1 Introduction

Accurate observations of atmospheric flow velocities, turbulence, and turbulence fluxes are critical for our understanding of all physical processes that occur in the atmospheric boundary layer and for the improvement of atmospheric modelling. Examples of intensely researched applications of turbulent fluxes include the closure of the surface energy balance (Foken, 2008), as well as the estimation of the carbon balance based on eddy-covariance observations, in which a very small systematic error can have a significant effect on the yearly carbon budget (Ibrom et al., 2007). Other applications include wind-power meteorology: turbulence is an important design parameter for wind turbines as the turbine loads are directly related to the velocity variances and turbulence measurements are therefore needed to find out whether a wind turbine can withstand the local flow conditions (Mücke et al., 2011; Dimitrov et al., 2015).

Our current understanding of atmospheric turbulence is, to a high degree, based on measurements performed with three-dimensional sonic anemometers deployed on meteorological towers. However, sonic anemometer measurements suffer from flow distortion due to the effects of both the structure(s) where the anemometer is mounted on, i.e., booms, clamps, and the bulk of the mast itself (e.g., Dyer, 1981; McCaffrey et al., 2017) and the anemometer itself. The latter effect has been recognized as a limitation for the accuracy of sonic anemometer observations for several decades (Wyngaard, 1981; Zhang et al., 1986; Grelle and Lindroth, 1994; van der Molen et al., 2004; Horst et al., 2015).

Some of the first wind-tunnel investigations on how the sonic anemometer structure impacts the measurements' accuracy were performed on the Kaijo-Denki DAT-300 sonic anemometer (Kraan and Oost, 1989; Mortensen, 1994). They showed



azimuth-dependent errors in the observed wind speed, which reflected the geometry of the probe head. These studies were followed by wind-tunnel investigations of the much more slender Gill R2 sonic anemometers by Grelle and Lindroth (1994), who showed the influence from the three supporting bars on the probe head leading to maximum wind speed errors of 15%, whereas the change of tilt within a small interval of angles showed less effect. This study was followed by that of Mortensen and Højstrup (1995), who showed influence on the accuracy of the measured velocity both from the ambient temperature and wind speed. Later, van der Molen et al. (2004) investigated Gill R2 and R3 sonic anemometers for a much wider range of tilt angles than those from the previous two studies. They demonstrated that the vertical velocity was severely underestimated at large tilt angles. Whereas surface sensible heat flux observations taken over forest increased by 4% using the calibration scheme by Grelle and Lindroth (1994), the calibration scheme by van der Molen et al. (2004) resulted in sensible heat flux increases of 15% for a different forested site. For the USA-1 (or its more modern version the uSonic3) sonic anemometer from Metek GmbH, Hamburg, Germany, two-dimensional and three-dimensional flow-distortion corrections were provided by Metek GmbH (2004) (hereafter M04). They are based on wind-tunnel observations for a number of azimuths and tilt angles.

Högström and Smedman (2004) documented an intercomparison between hot-film anemometers and Gill Solent R2 and R3 sonic anemometers. Both types of instruments were calibrated in a wind tunnel and subsequently intercompared in full-scale experiments. Whereas the hot-film anemometers retained their precision from the calibration, that of the sonic anemometers deteriorated in the field tests. Högström and Smedman (2004) argued that this difference could be explained by the effect of atmospheric turbulence and, hence, that wind-tunnel-based calibrations may therefore not be valid.

Another method for testing the precision and accuracy of sonic anemometers is to mount different brands closely and study the agreement between their turbulence measurements (e.g., Mauder et al., 2007; Kochendorfer et al., 2012). The challenge with this method is the difficulty to objectively determine which of the sonic anemometers measures best. Also, if agreement is found, this could be due to a similar error.

A third variant for assessing sonic anemometer performance is by comparing some of the same brand, by mounting them at different tilts and azimuths. Nakai and Shimoyama (2012) used five Wind-Master sonic anemometers mounted at different angles relative to each other and deduced flow distortion correction schemes based on the anemometers' different response as a function of both tilt and azimuth angles. Since the geometry of the Wind-Master is identical to that of the Solent R2 and R3, the resulting flow-distortion correction scheme could be compared to that of van der Molen et al. (2004). The new scheme by Nakai and Shimoyama (2012) pointed to slightly higher increases in the turbulent fluxes than that by van der Molen et al. (2004). Whereas this method avoids the potential problems associated with quasi-laminar wind-tunnel calibrations, the accuracy of the correction cannot be better than the accuracy of the instrument chosen as the reference. Also, the somewhat "busy" setup with several sonic anemometers in a small area can lead to additional and larger flow distortions than those using a single sonic anemometer.

Several combinations of the three different methods outlined above (wind-tunnel calibration, comparison of different brands of sonic anemometers, and tilting sonic anemometers of the same brand relative to each other) have also been demonstrated. Kochendorfer et al. (2012) used two sonic anemometers by R. M. Young and an orthogonal sonic anemometer by ATI as reference, and studied the observations of the vertical wind speed over a wide range of azimuth and tilt angles. They found





that for their sites, the vertical wind speed was underestimated by ≈11%, and when applying their derived corrections, the heat fluxes increased by 9–13%. A similar setup was used by Frank et al. (2013), who also showed that the CSAT3 sonic anemometer by Campbell, Logan, US, underestimated the vertical velocities and that this led to an underestimation of the sensible heat flux of about 10%. Horst et al. (2015) (hereafter H15) used a combination of all three methods to derive a flow distortion correction

for the CSAT3. Their correction, when applied to sensible heat flux data taken over an orchard canopy, showed a more modest effect closer to 5%. Based on the same data as those in Frank et al. (2013), Frank et al. (2016) demonstrated the use of a Bayesian model to estimate the most likely flow-distortion correction scheme of the CSAT3 and found a 10% increase in vertical velocities and sensible heat flux as well. Huq et al. (2017) presented a novel approach for estimating the accuracy of the CSAT3 by using numerical simulations. The results of the study pointed to flow-distortion errors of similar magnitude

as those in H15. The discrepancies in the findings of the previous studies foster the debate on the magnitude of the CSAT3 flow-distortion correction. Given the key role that sonic anemometers have in the field of experimental micrometeorology, it is of great importance to find objective standards by which accuracy and precision can be evaluated.

The aim of the current study is two-fold. First we introduce a new reference for evaluating sonic anemometer accuracy, and second, we evaluate the effect of flow -distortion corrections for two different sonic anemometers using this reference. The

two sonic anemometers are the USA-1, for which we apply the manufacturer's flow-distortion correction, which is based on wind-tunnel measurements, and the CSAT3, for which we apply the correction by H15. To our knowledge, the reference, which is simply the relation between the velocity spectra within the inertial subrange, has not been used previously for diagnosing sonic anemometer accuracy.

## 2 Background and methods

We first start by introducing the expected relations between velocity spectra within the inertial subrange in Sect. 2.1 and later introduce the flow corrections commonly used for sonic anemometers measurements in Sect. 2.2.

### 2.1 Inertial subrange

The inertial subrange corresponds to the region in the atmospheric energy spectrum where energy is neither produced nor dissipated, and where the transfer of energy from the energy containing range (buoyancy- and shear-produced energy) to the

dissipation range (kinetic to internal energy) is controlled by $\varepsilon$, which is the rate at which energy is converted to heat in the dissipation range (Kaimal and Finnigan, 1994).

Following the dimensional considerations of Kolmogorov (1941), the power spectrum of $u$, which is that of the along-wind component of the velocity, within the inertial subrange becomes

$$F_u(k_1) = \alpha \varepsilon^{2/3} k_1^{-5/3}, \tag{1}$$

where $k_1$ is the along-wind wavenumber and $\alpha$ is the universal Kolmogorov constant (≈0.5). Turbulence is locally isotropic within this range, which also means that all one-point correlations between velocity components become zero. Due to incom-




pressibility and isotropy, the velocity power spectra follows the relation,

$$F_u(k_1) = \frac{3}{4} F_v(k_1) = \frac{3}{4} F_w(k_1), \tag{2}$$

where $v$ and $w$ are the cross and vertical velocity components, respectively. Figure 1 illustrates idealized velocity spectra showing the spectral regions, the behavior of each velocity component, and the relations in the inertial subrange. It is important to note that Eqn. (2) is only an asymptotic relation valid for $1/L \ll k_1 \ll 1/\eta$ where $L$ is an outer scale of the turbulence, for example, the most energy containing scales, and $\eta$ is the Kolmogorov length scale.

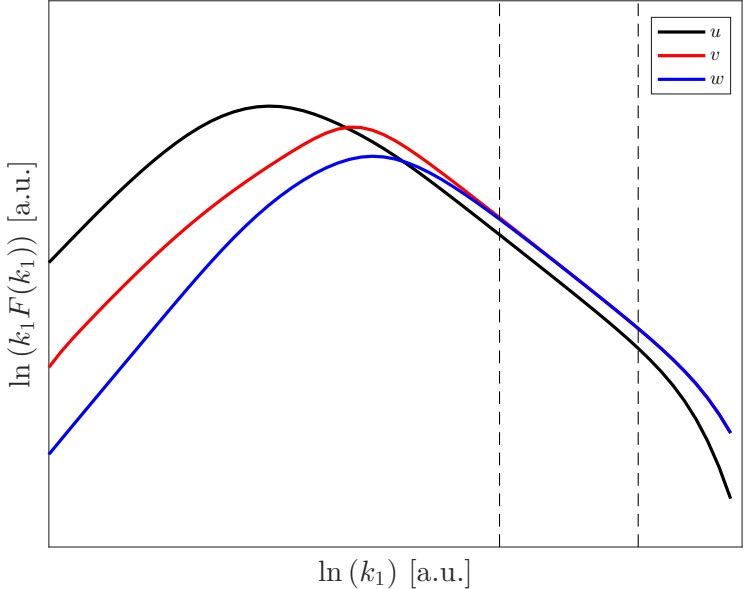

**Figure 1.** Idealized atmospheric velocity spectra showing the spectral regions and the relations in the inertial subrange (indicated within the vertical dashed lines)

## 2.2 Corrections to sonic anemometer measurements

### 2.2.1 Path-length averaging correction

For observations taken near the surface, or during stable atmospheric conditions, the distance between the sonic anemometer transducers (the path length $p$) over which the wind field is averaged may be a significant fraction of the length scale of the turbulence. A measured velocity power spectrum can therefore show a reduction of magnitude in the inertial subrange. Using similar methods as in Kaimal et al. (1968), Horst and Oncley (2006) (hereafter H06) calculated how path-length averaging influences sonic anemometer measurements for the geometries of the CSAT3 and Gill R3 sonic anemometers. The path-length averaging errors are expressed as transfer functions for each velocity component and depend on $k_1 p$. Here, we implement the results by H06 using the transfer functions for each of the velocity components by means of interpolation of tabular values





to observed $k_1p$ values. The tabular values for the CSAT3 are listed in H06, Table BI, Appendix B. Since the USA-1 has the same geometry as the Gill R3, the values in Table BII, Appendix B in H06 can be applied to the former instrument. It turns out that the effect of path-length averaging on the three velocity components is different for both the CSAT3 and Gill R3

geometries. For $k_1 < 1/p$, which is the most relevant range for this investigation, the $u$-component is more attenuated than the $v$- and $w$-components. This can crudely be understood by taking into account the incompressibility of the flow. For any given component $i$, the one-dimensional spectrum can be expressed by adding all the three-dimensional spectral densities with the same $k_1$:

$$F_i(k_1) = \int\limits_{-\infty}^{\infty}\!\!\int \Phi_{ii}(k_1,k_2,k_3)dk_2dk_3 \qquad (3)$$

where we follow the notation of H06. The majority of the energy for the $v$- and $w$-components comes in the inertial subrange from the three-dimensional spectral densities with the smallest magnitude of $\boldsymbol{k} = (k_1,k_2,k_3)$, i.e., when $k_2$ and $k_3$ are close to zero and $k \equiv |\boldsymbol{k}| \approx k_1$. However, for the $u$-component, $\Phi_{11}(k_1,0,0) = 0$ because of incompressibility, so the bulk of $F_1(k_1)$ comes from $\Phi_{11}(\boldsymbol{k})$ where $k_2$ or $k_3$ are far from zero. In these situations $k > k_1$, so the three-dimensional Fourier components contributing to the one-dimensional spectrum have shorter wavelengths. In general, shorter wavelengths mean that the effect

of averaging becomes stronger, thus we can expect the attenuation of the $u$-component to be strongest, at least for $k_1 < 1/p$.

### 2.2.2 Flow-distortion correction for the CSAT3 sonic anemometer

We implement the scheme by H15, which is based on that by Wyngaard and Zhang (1985) and calibrated through wind tunnel observations. The procedure follows:

1. Calculation of the length of the instantaneous wind vector $S = \sqrt{x^2 + y^2 + z^2}$, where $x$, $y$, and $z$ are the raw velocity
components in the instrument's coordinate system.

2. Projection of the velocity components $\mathbf{u} = (x,y,z)$ to the vectors defined by the paths of the sonic anemometer.

3. Calculation of the angle between the wind vector and each of the paths (subindex $p$), $\theta_i = \arccos(u_{p,i}/S)$, where $i = 1$–3 denote paths 1–3, and $u_{p,i}$ the projection of the velocity component on each path.

4. Correction (subindex $c$) of transducer shadowing $u_{p,i,c} = u_{p,i}/(0.84 + 0.16\sin\theta_i)$.

5. A final rotation of the corrected velocities back to a Cartesian coordinate system.

### 2.2.3 Flow distortion corrections for the Metek USA-1 sonic anemometer

There are two types of flow corrections available for the USA-1. The first one is a two-dimensional (2D) correction that takes into account the azimuth angle and, the second, a three-dimensional (3D) correction accounting for the tilt as well. Both are





suggested by M04. The 2D-corrected velocities are

$$x_{2D} = x\delta, \tag{4}$$
$$y_{2D} = y\delta, \tag{5}$$
$$z_{2D} = z + 0.031 U_r \left[\sin(3\alpha) - 1\right], \tag{6}$$

where $\delta = 1.00 + 0.015\sin(3\alpha + \pi/6)$, $U_r = \delta\left(x^2 + y^2\right)^{1/2}$, and $\alpha = -\text{atan2}(y, x)$. The 3D correction is applied through lookup tables (LUTs) derived from wind-tunnel measurements. Defining the horizontal and total velocity as $U = \left(x^2 + y^2\right)^{1/2}$ and $V = \left(x^2 + y^2 + z^2\right)^{1/2}$, respectively, and the azimuth and tilt angles as $\alpha = \text{atan2}(-y, -x)$ and $\phi = -\text{atan2}(z, U)$, the velocity, azimuth, and tilt are corrected as,

$$V_{3D} = n_c(\alpha, \phi)V, \tag{7}$$
$$\alpha_{3D} = \alpha + \alpha_c(\alpha, \phi), \tag{8}$$
$$\phi_{3D} = \phi + \phi_c(\alpha, \phi), \tag{9}$$

where $n_c(\alpha, \phi)$, $\alpha_c(\alpha, \phi)$, and $\phi_c(\alpha, \phi)$ are $\alpha$- and $\phi$-dependent correction factors[1], which are computed through Fourier series with coefficients $C_{f,i}(\phi)$ and $S_{f,i}(\phi)$ that are provided in the LUTs,

$$f_c(\alpha, \phi) = \sum_{i=0,3,6,9} \left[C_{f_c,i}(\phi)\cos(i\alpha) + S_{f_c,i}(\phi)\sin(i\alpha)\right], \tag{10}$$

where $f_c(\alpha, \phi)$ is either $n_c(\alpha, \phi)$, $\alpha_c(\alpha, \phi)$, or $\phi_c(\alpha, \phi)$. The LUTs are not given in M04 and so we provide them in Appendix A. The 3D-corrected velocities are,

$$x_{3D} = -V_{3D}\cos\alpha_{3D}\cos\phi_{3D}, \tag{11}$$
$$y_{3D} = -V_{3D}\sin\alpha_{3D}\cos\phi_{3D}, \tag{12}$$
$$z_{3D} = -V_{3D}\sin\phi_{3D}. \tag{13}$$

## 3 Sites and instrumentation

Measurements are collected from sonic anemometers mounted on three meteorological masts at two sites in Denmark: the Risø test site on the Zealand island and the Nørrekær Enge wind farm on northern Jutland (see Fig. 2). The Risø test site is over a slightly undulating terrain with a mix of cropland, grassland, artificial land, and coast (the Roskilde Fjord coastline is ≈250 m northwest of the turbine stands). The Nørrekær Enge wind farm is located ≈350 m southeast of the water body Limfjorden over flat terrain with a mix of croplands and grasslands.

At the Risø test site, a CSAT3 was mounted at 6.4 m above ground level (agl) on a 2.5-m boom on a 15-m tall tower. The boom was oriented 14° from the north. The tower was a triangular lattice structure with a side length of 0.4 m at the measurement height. The data acquisition unit was placed on the western leg of the tower, just below the boom.

---

[1]note that there is a typo in $V_{3D}$ in M04





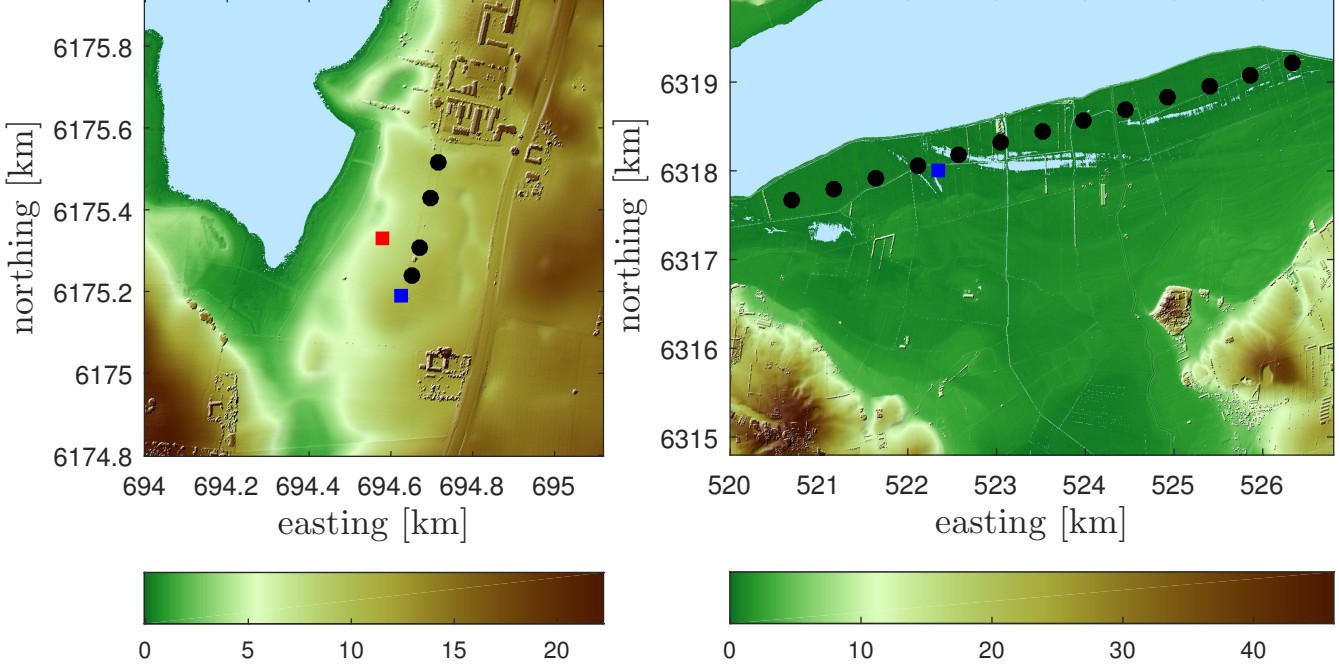

**Figure 2.** Locations of the sonic anemometer measurements. Wind turbines are indicated in black circles, masts with a CSAT3 on blue squares, and the mast with a USA-1 in a red square. The left panel shows the Risø test site and the right panel the Nørrekær wind farm site. The colorbar indicates the height above mean sea level in meters based on a digital surface elevation model (UTM32 WGS84)

Also at the Risø test site, but on a different mast, a USA-1 Basic was mounted at 16.5 m agl on a 2-m boom, which is oriented 15° from the north, on a 54-m tall tower that is located west of the wind turbine stands. The tower is a square lattice structure 0.3 m wide from bottom to top.

At the Nørrekær Enge wind farm, a CSAT3 was mounted at 76 m agl on a 3.1-m boom, which is oriented 192.5° from the north, on a 80-m mast that is located southeast of the row of wind turbines between stands 4 and 5 (numbered from left to right). The mast is an equilateral triangular lattice structure with a width of 0.4 m at 80 m.

5    At all sites the sonic anemometers were mounted so that their north was aligned with the boom direction. Thus, wind directions are hereafter relative to the sonic anemometer orientation where 0° is aligned with the boom. In Table 1, the specifications of the sonic anemometers at the two sites and the applied corrections are provided.

## 4   Data treatments

For all sonic anemometers, we analyze the time series of the three velocity components on a 10-min basis when $U > 3 \, \mathrm{m \, s^{-1}}$.
10   We apply azimuth and tilt rotations to the time series so that $u$ becomes aligned with the mean wind vector for each 10-min period. Finally, we compute all velocity spectra and co-spectra for each 10-min period.



**Table 1.** Sonic anemometer specifications for each measurement site including the use of flow distortion (FD) and/or path averaging (PA) corrections

| site | sonic anemometer | height agl [m] | $p$ [mm] | type of correction |
|---|---|---|---|---|
| Risø | USA-1 | 16.5 | 175 | PA (H06) and FD (M04) |
| Risø | CSAT3 | 6.4 | 115 | PA (H06) and FD (H15) |
| Nørrekær Enge | CSAT3 | 76.0 | 115 | FD (H15) |

### 4.1 USA-1 at the Risø test site

The USA-1 measurements at Risø were sampled at 20 Hz. We use 25401 10-min time series of measurements conducted in 2014 in order to have sufficient data covering all directions. We do all spectra calculations on both the raw (non-corrected data), the 2D- and 3D-corrected data. We also apply the path-length averaging correction by H06 to the 3D-corrected data.

### 4.2 CSAT3 at the Risø test site

The CSAT3 measurements at Risø were taken between November 2013 and mid-January 2014 and sampled at 60 Hz. For the analysis, it was required that all recorded velocities had a quality signal equal to zero. Two velocity corrections are performed, the path-length averaging (H06) and the flow-distortion correction suggested by H15. After filtering for no quality warnings, the amount of 10-min time series left are 2720.

### 4.3 CSAT3 at the Nørrekær Enge wind farm

The CSAT3 measurements at Nørrekær Enge were sampled at 10 Hz. We use 27837 10-min time series of measurements conducted in 2015, when no warnings were recorded by the CSAT3 and no precipitation was recorded by a rain gauge on the mast. We also apply the flow-distortion correction suggested by H15.

## 5 Results

For the three sonic anemometers, we show examples of velocity spectra ensemble-averaged over specific wind direction intervals. This is done to illustrate that within the chosen wavenumber range, the velocity spectra ratios approach the theoretical spectral slopes of the inertial subrange closely. We also show all 10-min velocity spectra ratios as function of direction (with and without flow corrections). For the specific case of the USA-1, we show the ratios of the velocity variances as function of direction as well.

### 5.1 USA-1 at the Risø test site

Figure 3 shows two examples of 3D-corrected velocity spectra, ensemble-averaged over two direction intervals for measurements of the USA-1 at the Risø test site. The spectra are normalized using the horizontal wind-speed magnitude, which is





found to be a better scaling factor than any velocity variance or co-variance in terms of data scatter, and are multiplied by $k_1^{2/3}$ so that the inertial subrange can be distinguished as a flat region. To such normalized spectra, we fit a 0-degree polynomial within a wavenumber range to show that within this range the spectra are indeed flat.

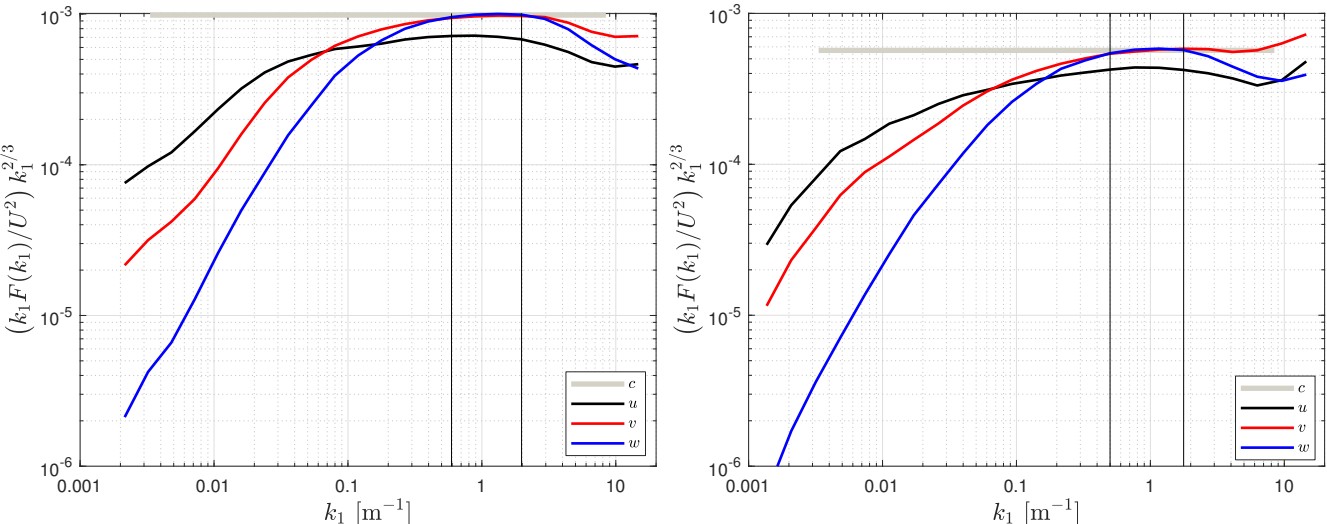

**Figure 3.** Ensemble-average 3D-corrected velocity spectra by the USA-1 at the Risø test site at 16.5 m for two directions intervals: one parallel (left panel) and another perpendicular (right panel) to the boom $\pm 10$ deg. A 0-degree polynomial ($c$) is fit to the wavenumber range indicated in black vertical lines

It is seen that for both direction intervals, the region in which the $w$-velocity spectrum becomes flat, is within the same wavenumber range ($0.5$ m$^{-1} \leq k_1 \leq 1.8$ m$^{-1}$). Thus, each 10-min spectrum can be analyzed within the same range, irrespective of the wind conditions. It is also observed that the spectra of the directions parallel to the sonic orientation have higher power spectral density than those of the directions perpendicular because for the latter the spectra are influenced by the fjord.

Figure 4-left shows the $w$- to $u$-spectra ratio for each non-corrected and 3D-corrected 10 min. It is clearly seen that the non-corrected data approach a ratio close to one, whereas the 3D-corrected data approach 4/3. Figure 4-right shows the 3D-corrected results for both the $w$- to $u$- and the $v$- to $u$-spectra ratios. It is also observed that for the latter the data also approach 4/3. $u$- and $v$-spectra do not change much after the 3D correction (not shown). Table 2 provides the computed velocity spectra ratios within the inertial subrange, for the direction interval where there is no direct influence by the mast, and for both the non-corrected measurements, the 3D corrected, and the path-length averaging- and 3D-corrected measurements.

Figure 5 shows the ratio of the 3D-corrected to the non-corrected velocity variances as function of wind direction. It is clearly seen that the 3D correction does not only change the spectral density of $w$ within the inertial subrange but that it increases the spectral density at all wavenumbers, and so the 3D-corrected variance is 4/3 the non-corrected one. As expected, the 3D correction does not change the $u$- and $v$-variances much.





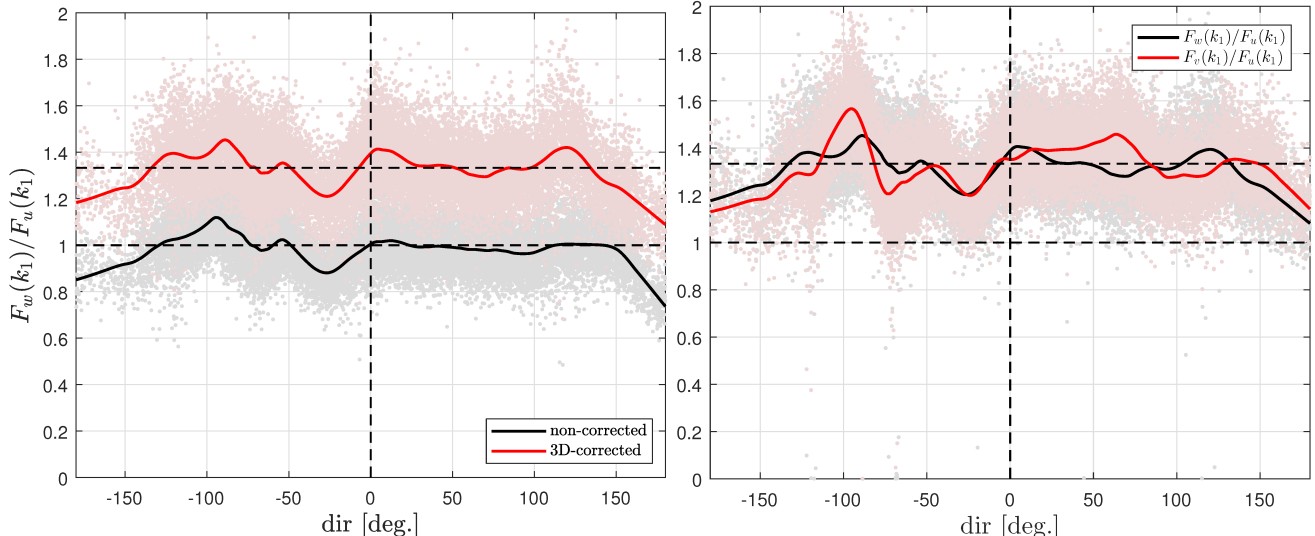

**Figure 4.** Velocity spectra ratios by the USA-1 at the Risø test site as function of wind direction. (Left frame) $w$- to $u$-velocity spectra ratio for the non- and 3D-corrected data. (Right frame) $w$- and $v$- to $u$-velocity spectra ratios for the 3D-corrected data. Each 10-min ratio is shown in markers and the solid lines show a moving average of the scatter. The dashed vertical line indicates the $0°$ direction and two dashed horizontal lines indicate the values 1 and 4/3

**Table 2.** Computed velocity spectra ratios within the inertial subrange for the direction range within $\pm120$ deg. The mean value is given $\pm$ one standard deviation.

| site | sonic anemometer | correction type | sharpened criteria | $F_w(k_1)/F_u(k_1)$ | $F_v(k_1)/F_u(k_1)$ |
|---|---|---|---|---|---|
| Risø | USA-1 | none | no | $0.999 \pm 0.097$ | $1.283 \pm 0.139$ |
| Risø | USA-1 | FD (M04) | no | $1.348 \pm 0.128$ | $1.345 \pm 0.145$ |
| Risø | USA-1 | FD (M04) and PA (H06) | no | $1.333 \pm 0.125$ | $1.332 \pm 0.145$ |
| Risø | CSAT3 | none | no | $1.132 \pm 0.065$ | $1.344 \pm 0.091$ |
| Risø | CSAT3 | FD (H15) | no | $1.194 \pm 0.070$ | $1.373 \pm 0.093$ |
| Risø | CSAT3 | FD (H15) and PA (H06) | no | $1.155 \pm 0.068$ | $1.320 \pm 0.089$ |
| Risø | CSAT3 | FD (H15) and PA (H06) | yes | $1.163 \pm 0.070$ | $1.314 \pm 0.087$ |
| Nørrekær Enge | CSAT3 | none | no | $1.061 \pm 0.217$ | $1.319 \pm 0.315$ |
| Nørrekær Enge | CSAT3 | FD (H15) | no | $1.117 \pm 0.234$ | $1.339 \pm 0.331$ |
| Nørrekær Enge | CSAT3 | FD (H15) | yes | $1.135 \pm 0.235$ | $1.317 \pm 0.178$ |





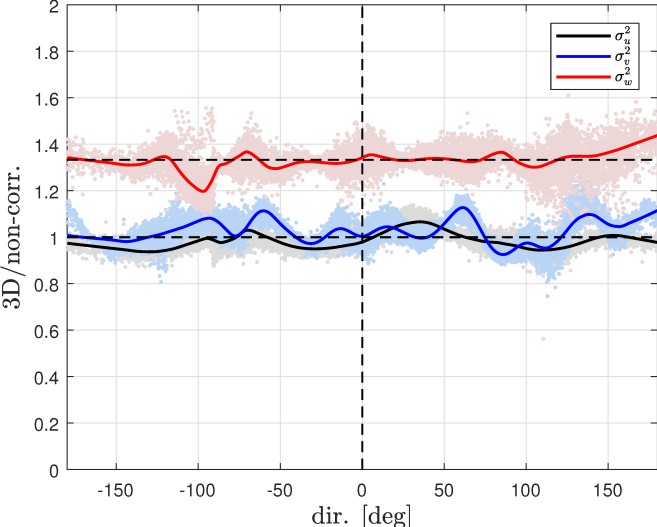

**Figure 5.** Ratios of the 3D-corrected to the non-corrected velocity variances by the USA-1 at the Risø test site as function of the wind direction. Each 10-min ratio is shown in markers and the lines show a moving average of the scatter.

## 5.2 CSAT3 at the Risø test site

From the investigated sonic anemometers, the CSAT3 at Risø had the lowest measurement height. Since the velocity spectra scale with height, the inertial subrange is expected to be within a range of higher wavenumbers compared to those from the other two sonic anemometers. The wavenumber range at which the premultiplied velocity spectra from this sonic anemometer shows an approximately flat range is $k_1 = [2,5]$ m$^{-1}$ (see Fig. 6). Such high wave numbers may be affected by white noise from the data acquisition itself. The upper limit of the $k_1$ interval chosen for analysis is therefore limited, particularly for the $u$ and $v$ components (refer to Appendix B for an explanation of why each velocity component is affected differently by noise), which causes the spectral slope to be greater than $-5/3$.

For the selected wavenumber range, the ratios are computed for each 10-min sample. Figure 7 illustrates the computed velocity-component spectra ratios. Both $F_w(k_1)/F_u(k_1)$ and $F_v(k_1)/F_u(k_1)$ show very low values for absolute directions greater than $\approx 150$ deg. For lower absolute directions, the ratios vary between 1.0 and 1.6 (Fig 7-left). For most of the directional intervals, the ratio $F_v(k_1)/F_u(k_1)$ is clearly higher than the $F_w(k_1)/F_u(k_1)$ ratio. To understand whether the data spread is due to incorrect choice of spectral range, in which perfect inertial subrange behavior cannot be expected, we sharpen the selection criteria, i.e., we try to filter out 'poor' spectra, by assuring that within the selected wavenumber range both the spectral slope is $-5/3 \pm 0.003$ and $|F_{uw}/\sqrt{F_u F_w}| < 0.02$ (i.e., narrowing for isotropy). However, as shown in Table 2, the results for the mean velocity ratios are insensitive to the poor spectra filter.

In Fig. 7-right, we show the mean of the spectra ratio over each 5° interval for three cases: no correction, H15 correction, and combining the H06 and H15 corrections. Whereas the H15 correction increases the ratios, by adding the H06 correction the





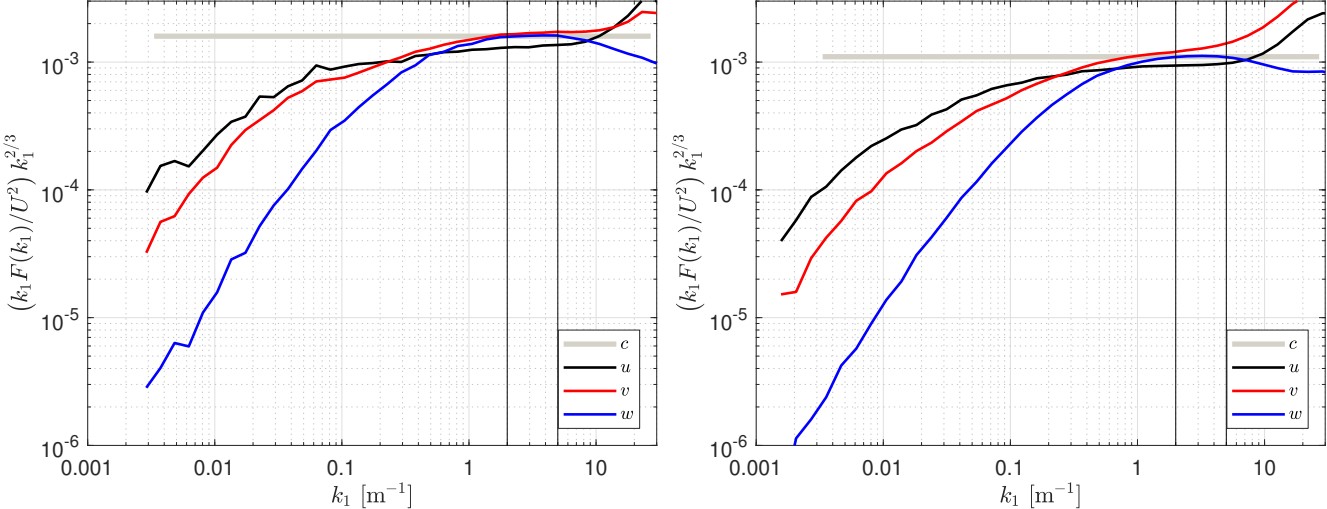

**Figure 6.** Similar to to Fig. 3, but for the CSAT3 at the Risø test site. For the direction parallel to the boom $\pm 10°$ (left frame), the average spectra is computed over 72 10-min samples, whereas for the directions perpendicular to the boom $\pm 10°$ (right frame), the average is based on 453 10-min samples

ratio is reduced. It can be observed that the effect of path-length averaging (H06) is opposite to that of transducer shadowing (H15). As discussed before, $F_u$ is attenuated more than $F_v$ and $F_w$ by path-length averaging in the inertial subrange. Therefore,
10 when path-length averaging is accounted for, the ratios reduce.

### 5.3 CSAT3 at the Nørrekær Enge wind farm

Figure 8 shows two examples of normalized velocity spectra, ensemble-averaged over two direction intervals for measurements of the CSAT3 at Nørrekær Enge as well as the polynomial fit within a chosen wavenumber range. Due to noise at the highest wavenumbers ($k_1 > 1 \text{ m}^{-1}$), we limit the range to a close to noise-free wavenumber. Similar to the velocity spectra measured
15 by the CSAT3 at the Risø test site, the $w$ spectrum follows closely the $u$ spectrum and the $v$ spectrum shows the highest spectral density within the inertial subrange ($0.38 \text{ m}^{-1} \leq k_1 \leq 0.88 \text{ m}^{-1}$).

Figure 9 shows the $w$ and $v$ to $u$ spectra ratios for each 10 min. The result is very similar to that for the CSAT3 at the Risø test site where within a range of directions of $\pm 150°$, the $w$ to $u$ spectra ratios are close to one, whereas the $v$ to $u$ spectra ratios are close to 4/3. The boom/mast structure has a greater effect on the CSAT3 at the Nørrekær Enge wind farm than at the
5 Risø site as expected due to the setup. For both sites, the effect of the boom/mast at directions close to $\pm 180°$ is very similar. In agreement with the findings using the CSAT3 at Risø (Sect. 5.2), the H15 correction increases both the $F_w(k_1)/F_u(k_1)$ and $F_v(k_1)/F_u(k_1)$ spectra ratios (particularly for the former) but not enough to reach the 4/3 value for $F_w(k_1)/F_u(k_1)$.





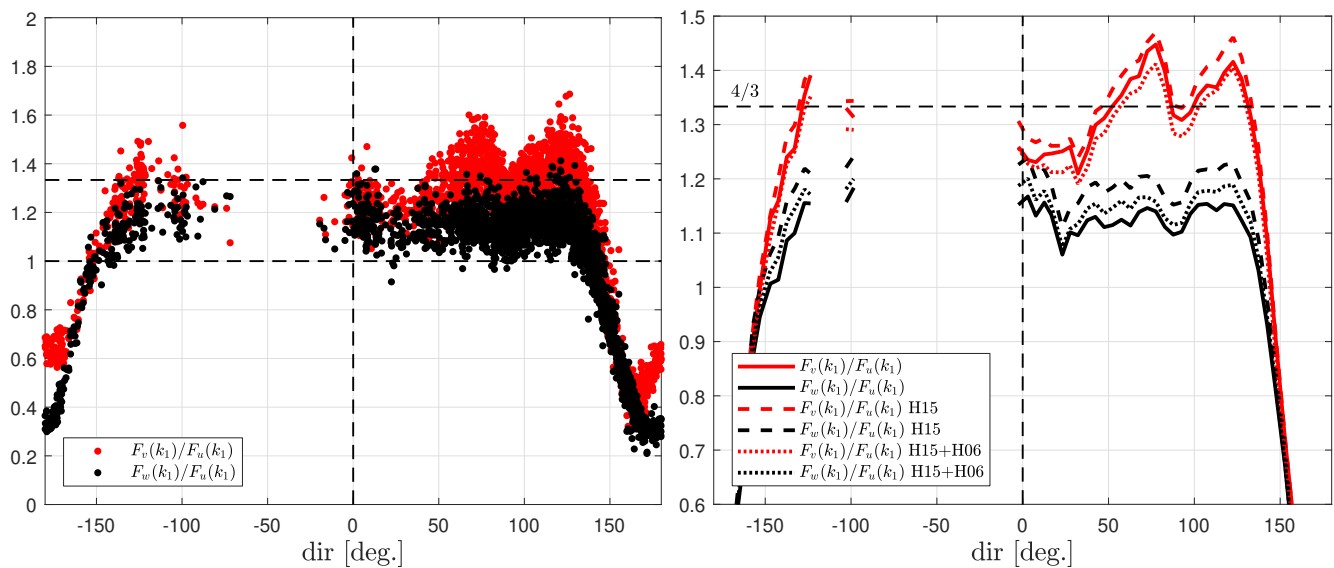

**Figure 7.** Velocity spectra ratios by the CSAT3 at the Risø test site as a function of wind direction for each 10-min period (left frame) and averaged over five degree intervals (right frame).

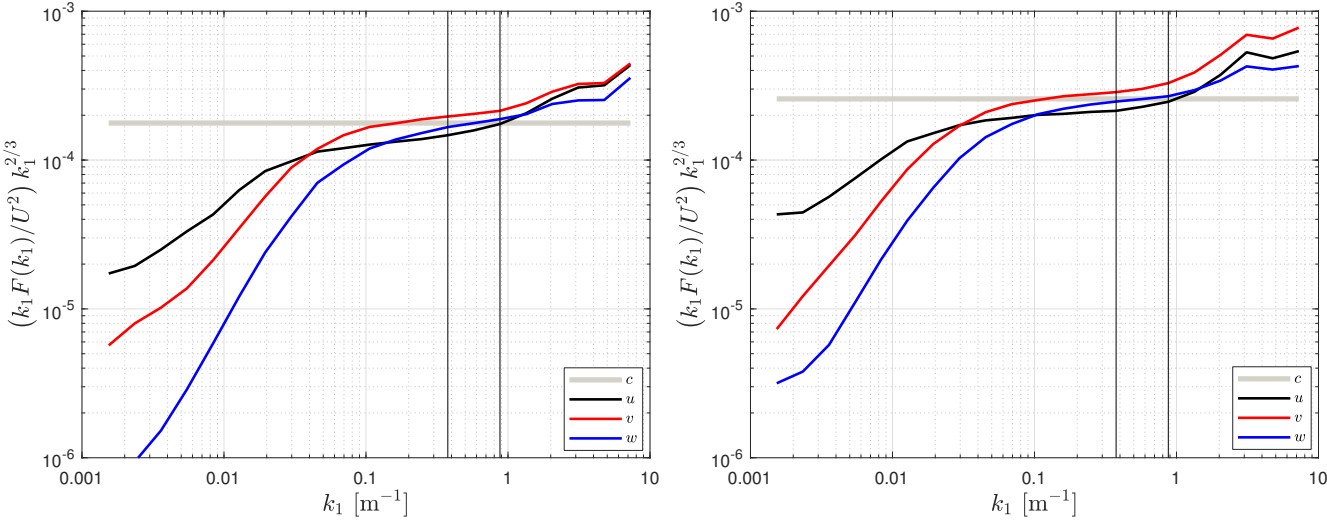

**Figure 8.** Similar to Fig. 3 but for the CSAT3 at the Nørrekær Enge wind farm





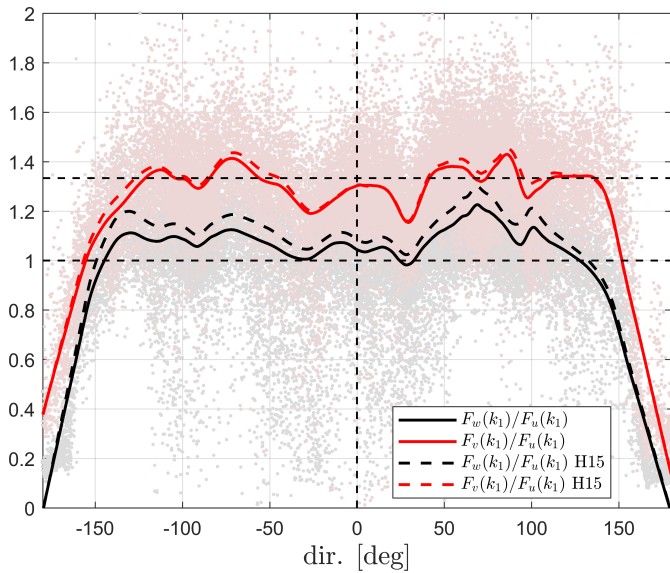

**Figure 9.** CSAT3 velocity spectra ratios with wind direction at the Nørrekær Enge wind farm

## 6    Discussion

### 6.1    Uncertainties

The aim of the spectral analysis displayed in Figs. 3, 6, and 8 was to find the optimal inertial subrange for each site and setup. A high-end limitation to this interval can be the presence of white noise in the spectra, which would tend to reduce the examined spectral ratios. For the velocity spectra at all three locations, we observe that the high frequency $w$-noise is the lowest of the three velocity components and is proportionally lower for the CSAT3 than for the USA-1, which is consistent with its larger path elevation angle as explained theoretically in Appendix B. According to the theory, the noise in the $v$- and $u$-spectra should

be identical irrespective of the wind direction relative to the boom. The data showed deviations from this prediction. In addition, for the Risø CSAT3 setup, numerous tests with regards to both wavenumber and frequency ranges were tested, resulting in only very slight changes to the results in Fig. 7 (not shown). Another test for the robustness of the results was performed by selecting only those spectra, which showed close to perfect inertial subrange behavior within the selected wavenumber range (a close to $-5/3$ slope for $F_w(k_1)$ and a low co-covariance, see "sharpened criteria" in Table 2), which affected the ratio $\approx$0.5-1.5% only.

Another potential source of error comes from the choice of coordinate system in which the spectra are calculated. Here, we used two rotations for each 10-min block of data, whereas Horst et al. (2015) used the planar-fit coordinate system by Wilczak et al. (2001). We tested whether an error in rotation angle would change the results. This was done by rotating the sonic and

5    applying an isotropic inertial subrange 3D spectral velocity tensor, as in H06, to calculate the nominal component spectra for this configuration. A change of up to $\pm5°$ in the rotation angles of the sonic anemometer about vertical and transverse axes resulted in a less than 0.7% reduction in the spectral ratio; therefore, we consider rotation-related errors to be of no importance.



## 6.2 Implications

We base our analysis on theoretical arguments about the $w$- and $v$- to $u$-velocity spectral ratios, which should be equal to
4/3 within the inertial subrange. We find such ratios by applying the 3D wind-tunnel-derived flow-distortion corrections to
atmospheric velocity measurements performed with a USA-1 (Table 2); whereas applying a flow distortion correction to the
CSAT3 results in ratios within the range 1.12–1.19. Assuming that this discrepancy is due to remaining uncorrected flow
distortion and further by assuming that flow distortion affects the frequencies equally, which was supported by Huq et al.
(2017), the imperfect ratios correspond directly to underestimations in velocity variances. Hence, the results in Table 2 can be
used to correct the CSAT3's $F_w(k_1)/F_u(k_1)$ ratio to 4/3. If we assume that the error is only affecting the $w$ component, we can
estimate the correction by dividing the ideal ratio with the observed ratio (Table 2). Using the observed CSAT3 ratios for Risø
and Nørrekær Enge without any correction applied, the $w$ variance should then increase by 18–26%, which means that the $w$
component itself should increase by 8–12%. These error ranges are in agreement with the results by Frank et al. (2016), and
significantly higher than those suggested by Huq et al. (2017). A drawback with the method is, however, that it is not possible
to determine which of the velocity components is affected. The ideal ratio can also be achieved, e.g., by assuming an 4–6%
error on all velocity components (positive for $u$ and $v$, and negative for $w$).

Another clear result from the presented analyses concerns the difference between observed mast/boom/instrument shadowing
for the USA-1 and CSAT3; even from narrow masts and relatively long supporting booms, the mast influence is more marked
for the CSAT3 than for the USA-1. Whereas Foken (2008) recommended the use of sonic anemometers without a pole directly
under the sonic measurement volume for atmospheric turbulence research, we here stress that this statement can at best be valid
only for a limited wind direction interval. For anemometers mounted on bulky walk-up towers, the direction interval where data
will be biased from the tower will likely be much larger. We further stress that a sonic anemometer that cannot reproduce a 4/3
ratio in the inertial subrange cannot be trusted to give accurate observations of all velocity components. Despite a higher ratio
of transducer diameter to path length, which is sometimes used as a sonic anemometer quality marker, the USA-1, including
the wind-tunnel-derived flow distortion correction, therefore comes out better from our analysis.

## 6.3 References in sonic anemometry quality assessments

We suggest that the spectral ratios of velocity components within the inertial subrange are a valuable addition to field tests
and wind-tunnel calibrations. The advantage of the presented method is that any sonic anemometer can be tested provided that
inertial subrange characteristics are expected from the particular measurements. Unlike sonic anemometer intercomparisons,
where ideal flat and uniform sites are preferred (e.g. Mauder et al., 2007), the spectral ratio method did not seem to be sensitive
to the spatial heterogeneity at the sites used here. As mentioned above, a limitation to our method is that the accuracy of
individual velocity components cannot be assessed; the observations from the USA-1, although almost perfect in terms of
the 4/3 ratio, might still be inaccurate if all three velocity components are biased. Looking ahead, a further reference for
sonic anemometer measurements could be found in small-volume lidar anemometry (Abari et al., 2015), which is free of flow
distortion.



## 6.4 Can wind-tunnel based calibrations be trusted in atmospheric turbulence?

Starting with Högström and Smedman (2004), the validity of wind-tunnel calibrations for sonic anemometer has been questioned for applications in the turbulent atmosphere. Using large-eddy simulation results, Huq et al. (2017) argued that the magnitude of flow-distortion error caused by the sonic anemometer is smaller under turbulent conditions than under quasi-laminar flow, while also showing that the flow-distortion error does not depend on the frequency of the fluctuations. Taken the latter result to the extreme low-frequency limit, these two results appear inconsistent. In this study, the application of a flow-

distortion correction for the USA-1, derived from wind-tunnel observations, led to near-perfect spectral ratios in the inertial subrange, whereas that by H15, based on both field tests and wind-tunnel observations, did not. Provided that the wind-tunnel reference instrument is accurate and the blockage ratio in the tunnel is small, we argue that flow distortion can be correctly quantified also in quasi-laminar flow, because the turbulence eddy sizes are significantly larger than the transducer size. In this way, the atmospheric turbulent flow appears laminar as seen from the transducer. An explanation for the deviation of the results

between sonic anemometer observations in wind tunnel and field tests in Högström and Smedman (2004) could also be that the velocities recorded by the early Gill sonic anemometers showed a marked temperature dependence (Mortensen and Højstrup, 1995).

## 7 Conclusions

The accuracy of atmospheric turbulence measurements performed by sonic anemometers was investigated using two instru-

25 ments, a CSAT3 and a USA-1, at two locations in Denmark. This was achieved by computing velocity spectra ratios within the inertial subrange. We propose to perform such an analysis, in addition to field site intercomparisons and wind-tunnel calibrations, as a method to assess the accuracy of sonic anemometer measurements. It was found that 3D flow corrections applied to measurements from the USA-1 helped recovering the 4/3 ratio of the $w$- to the $u$-velocity spectra that is expected within the inertial subrange. The 3D corrections also have a strong influence on the estimated $w$-variances, which are systematically found to be $\approx$4/3 of the uncorrected values. For the CSAT3, which is commonly categorized as the sonic anemometer closest to be a distortion-free instrument, the ratio of the $w$- to the $u$-velocity spectra is $\approx$1.1 without applying a flow-distortion correction.

5 Using a previously proposed flow-distortion correction, the ratios changed to $\approx$1.15 on average, pointing to that more work is needed to correctly quantify the flow distortion of this instrument. We also found that the influence of the mast, boom, and the instrument itself was higher on the CSAT3 compared to the USA-1 measurements.

*Data availability.* Sonic anemometer data are available under request to AP

## Appendix A: Metek USA-1 3D flow-distortion corrections



**Table A1.** LUT for $\alpha_c(\alpha, \phi)$

| $\phi$ [deg] | $C_{\alpha_c,0}$ | $C_{\alpha_c,3}$ | $S_{\alpha_c,3}$ | $C_{\alpha_c,6}$ | $S_{\alpha_c,6}$ | $C_{\alpha_c,9}$ | $S_{\alpha_c,9}$ |
|---|---|---|---|---|---|---|---|
| -50 | -10.7681 | 1.83694 | 8.12521 | 1.76476 | -0.120656 | -0.31818 | 1.30896 |
| -45 | -7.57048 | 2.25939 | 4.22328 | -0.0394204 | -0.112215 | -0.289935 | 1.99387 |
| -40 | -6.77725 | 0.293479 | 3.05333 | -1.16341 | 0.433886 | 0.207458 | 1.05195 |
| -35 | -4.12528 | 2.24741 | 0.286582 | -0.936084 | 0.205636 | -0.399336 | 1.57736 |
| -30 | -2.00728 | 3.63124 | -0.325198 | -0.821254 | 0.236536 | -0.303478 | 0.854497 |
| -25 | -3.1161 | 3.91749 | -0.682098 | -0.274558 | 0.401386 | -0.531782 | 0.470723 |
| -20 | -1.73949 | 3.5685 | -0.253107 | 0.0306742 | 0.236975 | -0.290767 | -0.224723 |
| -15 | -2.59966 | 2.7604 | -0.425346 | 0.0557135 | 0.0392047 | 0.222439 | -0.364683 |
| -10 | -1.80055 | 2.02108 | -0.259729 | 0.161799 | 0.117651 | 0.513197 | -0.0546757 |
| -5 | -1.02146 | 1.22626 | -0.469781 | -0.177656 | 0.402977 | 0.408776 | 0.513465 |
| 0 | 0.152354 | 0.208574 | 0.051986 | -0.102825 | 0.480597 | -0.0710578 | 0.354821 |
| 5 | 0.310938 | -0.703761 | -0.0131663 | 0.0877815 | 0.546872 | -0.342846 | 0.176681 |
| 10 | 0.530836 | -1.68132 | -0.0487515 | 0.0553666 | 0.524018 | -0.426562 | -0.0908979 |
| 15 | 1.70881 | -2.46858 | -0.487399 | 0.207364 | 0.638065 | -0.458377 | -0.230826 |
| 20 | 2.38137 | -3.37747 | 0.026278 | 0.0749961 | 0.759096 | 0.105791 | 0.0287425 |
| 25 | 3.81688 | -4.13918 | -0.690113 | 0.170455 | 0.474636 | 0.424845 | 0.232194 |
| 30 | 3.49414 | -3.82687 | -0.229292 | 0.54375 | 0.322097 | 0.387805 | 0.823967 |
| 35 | 4.1365 | -3.22485 | 0.752425 | 0.755442 | 0.623119 | 0.250988 | 1.26713 |
| 40 | 5.04661 | -2.53708 | 1.23398 | 0.623328 | 0.653175 | -0.359131 | 1.43131 |
| 45 | 4.26165 | -3.12817 | 2.61556 | 0.0450348 | -0.330568 | -0.34354 | 0.81789 |





**Table A2.** LUT for $\phi_c(\alpha, \phi)$

| $\phi$ [deg] | $C_{\phi_c,0}$ | $C_{\phi_c,3}$ | $S_{\phi_c,3}$ | $C_{\phi_c,6}$ | $S_{\phi_c,6}$ | $C_{\phi_c,9}$ | $S_{\phi_c,9}$ |
|---|---|---|---|---|---|---|---|
| -50 | 5.77441 | -2.19044 | 0.123475 | -0.229181 | 0.226335 | 0.271943 | 0.0434668 |
| -45 | 3.82023 | -1.6847 | 0.315654 | 0.562738 | 0.175507 | -0.0552129 | -0.110839 |
| -40 | 2.29783 | -1.04802 | 0.0261005 | 0.239236 | 0.125053 | -0.310631 | 0.388716 |
| -35 | 1.37922 | -1.0435 | 0.302416 | -0.0112228 | 0.333846 | -0.459678 | 0.172019 |
| -30 | 0.837231 | -0.593247 | -0.199916 | -0.0591118 | 0.19883 | -0.307377 | 0.182622 |
| -25 | -0.0588021 | -0.0720115 | -0.6826 | -0.253726 | 0.348259 | -0.322761 | 0.0059973 |
| -20 | -0.0333721 | 0.101664 | -1.41617 | -0.136743 | 0.332169 | -0.244186 | -0.0612597 |
| -15 | 0.0423739 | 0.0428399 | -1.90137 | -0.187419 | 0.148025 | 0.06782 | -0.0317571 |
| -10 | 0.318212 | 0.126425 | -2.07763 | -0.0341571 | 0.198621 | 0.178598 | 0.103543 |
| -5 | 0.721731 | -0.0274247 | -2.10221 | -0.081822 | 0.36773 | 0.0848013 | 0.184226 |
| 0 | 1.65254 | -0.0582368 | -2.18993 | -0.0802346 | 0.234886 | -0.0545883 | -0.0092531 |
| 5 | 2.49129 | -0.116475 | -2.11283 | 0.112364 | 0.247405 | -0.115218 | -0.0682998 |
| 10 | 2.99839 | -0.0867988 | -2.04382 | 0.219581 | 0.207231 | -0.0981521 | -0.0581594 |
| 15 | 3.55129 | -0.160112 | -1.8474 | 0.22217 | 0.2794 | -0.0323565 | -0.0951596 |
| 20 | 3.20977 | -0.137282 | -0.966014 | 0.183032 | 0.380154 | 0.155093 | -0.0557369 |
| 25 | 3.38556 | -0.0596863 | -0.898053 | 0.20526 | 0.39357 | 0.421141 | -0.00842409 |
| 30 | 3.18846 | 0.266264 | -0.0951907 | 0.166895 | 0.373018 | 0.338146 | 0.187917 |
| 35 | 2.60134 | 0.442007 | 0.211612 | -0.114323 | 0.359926 | 0.224424 | 0.209482 |
| 40 | 2.04655 | 1.08915 | 0.470385 | -0.333096 | 0.268349 | 0.263547 | 0.264963 |
| 45 | 0.987659 | 1.54127 | 0.815214 | -0.504021 | -0.0835985 | 0.197387 | 0.0819912 |



**Table A3.** LUT for $n_c(\alpha, \phi)$

| $\phi$ [deg] | $C_{n_c,0}$ | $C_{n_c,3}$ | $S_{n_c,3}$ | $C_{n_c,6}$ | $S_{n_c,6}$ | $C_{n_c,9}$ | $S_{n_c,9}$ |
|---|---|---|---|---|---|---|---|
| -50 | 1.23095 | -0.0859199 | -0.0674271 | 0.0160088 | 0.0363397 | 0.0141701 | -0.0271955 |
| -45 | 1.19323 | -0.0430575 | 0.00309311 | 0.0430652 | 0.0225135 | 0.000740028 | -0.0114045 |
| -40 | 1.17255 | -0.0206394 | 0.0145473 | 0.0399041 | -0.00592748 | -0.00650942 | -0.00762305 |
| -35 | 1.15408 | -0.00768472 | 0.0614486 | 0.0382888 | 0.0123096 | -0.0124673 | -0.00598534 |
| -30 | 1.12616 | 0.00000536 | 0.0636543 | 0.0386879 | 0.0153428 | -0.014148 | -0.000210096 |
| -25 | 1.09976 | 0.00667086 | 0.0705414 | 0.0198549 | 0.0165582 | -0.0114517 | -0.00115495 |
| -20 | 1.07518 | 0.00583915 | 0.0591098 | 0.011127 | 0.0104259 | -0.00665653 | 0.00119842 |
| -15 | 1.05173 | 0.00731099 | 0.0527018 | 0.00230123 | 0.00587927 | -0.00229463 | -0.00297294 |
| -10 | 1.02428 | 0.00885121 | 0.0330304 | -0.000597029 | 0.00340367 | -0.000745781 | -0.000283634 |
| -5 | 1.011 | 0.00930375 | 0.0218448 | -0.0046575 | 0.00203972 | -0.00112652 | 0.00179908 |
| 0 | 1.00672 | 0.0105659 | 0.0034918 | -0.00844128 | 0.00228384 | -0.000824805 | 0.000200667 |
| 5 | 1.01053 | 0.00885115 | -0.0182222 | -0.00894106 | -0.000719837 | -0.000420398 | -0.00049521 |
| 10 | 1.02332 | 0.00618183 | -0.035471 | -0.00455248 | -0.00215202 | -0.00229836 | -0.000309162 |
| 15 | 1.04358 | 0.00648413 | -0.0494223 | 0.000323015 | -0.00396036 | -0.00465476 | -0.000117245 |
| 20 | 1.06928 | 0.00733521 | -0.0638425 | 0.0101036 | -0.00829634 | -0.0073708 | -0.00051887 |
| 25 | 1.09029 | 0.00396333 | -0.0647836 | 0.0187147 | -0.0126355 | -0.0115659 | 0.000482614 |
| 30 | 1.11877 | 0.00299473 | -0.0661552 | 0.0293485 | -0.00957493 | -0.00963845 | 0.0029231 |
| 35 | 1.13779 | 0.00812517 | -0.0526322 | 0.0341525 | -0.00971735 | -0.0114763 | 0.0013481 |
| 40 | 1.16659 | -0.00869651 | -0.0537855 | 0.0290825 | -9,89E+00 | -0.0133731 | 0.0117738 |
| 45 | 1.18695 | -0.0289647 | -0.0461693 | 0.030231 | -0.0121524 | -0.00667729 | 0.00565286 |





## 10  Appendix B:  Sonic anemometer noise

The transformation matrix to convert the three sonic path velocities $\boldsymbol{s} = (s_1, s_2, s_3)$, which are assumed positive from the lower to the upper acoustical transducer, to right-handed orthogonal velocity components $\boldsymbol{u} = (u_1, u_2, u_3) = (u, v, w)$ with $u$ in the direction of the horizontal boom, $v$ horizontal and transverse to $u$, and $w$ vertical and positive upwards, is

$$T = \begin{pmatrix} -\frac{2\sec\phi_p}{3} & \frac{\sec\phi_p}{3} & \frac{\sec\phi_p}{3} \\ 0 & \frac{\sec\phi_p}{\sqrt{3}} & -\frac{\sec\phi_p}{\sqrt{3}} \\ \frac{\csc\phi_p}{3} & \frac{\csc\phi_p}{3} & \frac{\csc\phi_p}{3} \end{pmatrix} , \tag{B1}$$

15  where $\phi_p$ is the path elevation angle, so

$$u_i = T_{ij}s_j, \tag{B2}$$

and we also assume the sonic anemometer paths to be oriented in the azimuthal direction like the CSAT3 or the USA-1.

Suppose now that the sonic anemometer signals are composed of uncorrelated, white noise $\langle s_i s_j \rangle = \sigma_s^2 \delta_{ij}$, where $\delta$ is the Kronecker delta symbol and $\sigma_s^2$ is the noise variance. The resulting noise on the orthogonal velocity components then becomes

20  $$\langle u_i u_j \rangle = \langle T_{ik}s_k T_{jl}s_l \rangle = \sigma_s^2 T_{ik}\delta_{kl}T_{jl} = \sigma_s^2 T_{ik}T_{jk}$$

$$= \sigma_s^2 \begin{pmatrix} \frac{2\sec^2\phi_p}{3} & 0 & 0 \\ 0 & \frac{2\sec^2\phi_p}{3} & 0 \\ 0 & 0 & \frac{\csc^2\phi_p}{3} \end{pmatrix} \tag{B3}$$

Since the $u$- and $v$-components behave identically in terms of noise, the error is also given by Eqn. (B3) if the components are rotated into the mean wind direction coordinate system and as long as the wind vector is horizontal. Also, since the noise is assumed white, the relative strengths of noise-dominated spectra will also follow Eqn (B3). The ratio between the horizontal and vertical spectra will therefore increase rapidly with path elevation angle as shown in Fig. B1. Unit ratio occurs for $\phi_p = \tan^{-1}(2^{-1/2}) \approx 35°$ or at a path zenith angle of $90° - \phi_p \approx 55°$. This is also the path elevation angle where the sum of the three component variances obtains a minimum of exactly three times $\sigma_s^2$. Because of flow distortion, sonic anemometers do not have such a low path elevation angle.

*Author contributions.*  AP analyzed the USA-1 measurements at the Risø test site and the CSAT3 measurements at the Nørrekær wind farm. ED analyzed the CSAT3 measurements at the Risø test site. AP and ED contributed equally to the preparation of the paper. JM helped in the analysis and interpretation of the data and revised the contents of the manuscript. ED and JM came up with the idea of using the velocity spectra to diagnose flow-distortion effects on the sonic anemometer measurements

5  *Competing interests.*  The authors declare that they have no conflict of interest





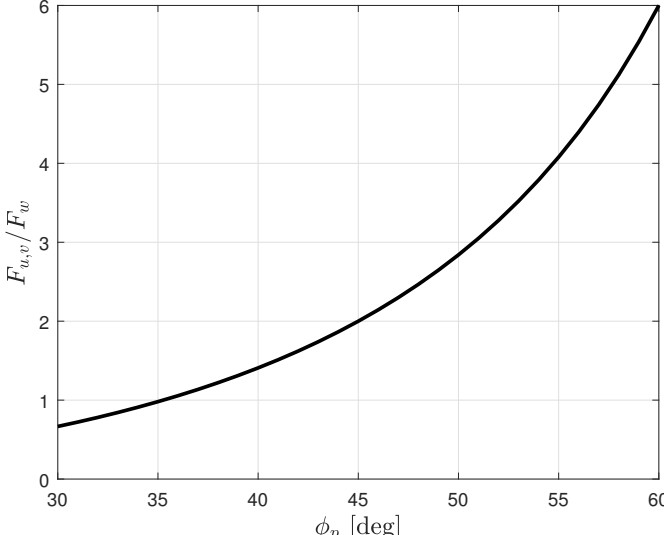

**Figure B1.** The ratio of the noise level in the horizontal velocity components to the vertical one as a function of path elevation angle

*Acknowledgements.* We would like to thank the technical support of the Test and Measurements section at DTU Wind Energy and in particular to Søren W. Lund. ED would like to thank the TrueWind project, which is funded by the Energy Technology Development and Demonstration Program (EUDP), Denmark for financial support. Finally, we would like to thank Tom Horst for many inspiring discussions on flow distortion corrections.





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
