# Peer review of "A method to assess the accuracy of sonic anemometer measurements"

_Atmospheric Measurement Techniques, 2018_

## Referee Comment (RC1) · Anonymous Referee #2 · 22 Oct 2018

In this paper, the authors present a novel methodology to evaluate the relative accuracy of u, v, and w measurements from a sonic anemometer by applying Kolmogorov theory to the relative magnitude of the u, v, and w spectra within the inertial subrange. Based on that theory, the v and w spectra should be 4/3 the magnitude of the u spectra. Using field data from Metek USA-1 and CSAT3 sonic anemometers at different towers, different field sites, without any shadowing correction, with shadowing correction, and with path averaging correction, the 4/3 relationship was tested. For the Metek, while the uncorrected anemometer was much lower than 4/3 for the w-to-u relationship, after applying a wind tunnel based calibration provided by the manufacturer this relationship becomes ∼4/3. For the uncorrected CSAT3, the w-to-u relationship is a little close to 4/3 than was the case for the uncorrected Metek, but after applying a shadowing

correction, the ratio is still lower than 4/3, being ~1.2. Thus, the CSAT3 correction could be interpreted as providing only a partial amount of the correction required to achieve a 4/3 relationship.

This is a very novel idea, and has the potential to be fairly influential in the discipline. There are two items that seem very important that should be given more emphasis. First, this technique is among the few that does not require a comparison between one anemometer and another. Rather, it can be applied for any single anemometer at more-or-less any field site (I recognize that this is mentioned in the paper, but this is EXTRMEMELY important, so make sure there is no doubt the reader appreciates how powerful this statement is). Second, because this methodology is entirely based on Kolmogorov theory about the 4/3 ration between w-u and v-u spectra, it should be emphasized that in general all of the results from the v-u tests conform to this theory. While the reader must evaluate results such as Table 2 to determine whether w-u ratios of ~1.0, ~1.1, ~1.2 are evidence of underestimated w measurements, it is crucial to note that the v-u ratios are almost entirely 4/3 for all cases. This gives a lot of credibility that the theory of isotropy is correct, and that the 4/3 standard is reasonable.

One improvement that will be necessary is to better clarify the differences between 4/3, 5/3, and 2/3 slopes in the inertial subrange. I was completely confused upon my first reading, and I had to consult my Kaimal and Finnigan (1994) book to sort this out. While it is clear from Equation 1 that the slope of the inertial subrange is -5/3, figure 1 is a -2/3 slope, which presumable is because it is a "frequency weighted" spectra where the y-axis is actually spectra multiplied by wave number. This is related to second sentence in section 5.1, where the frequency-weighted spectra is multiplied by $k^{(2/3)}$ to give a straight line. This is entirely confusing unless the reader recognizes that it is already multiplied by $k^{(3/3)}$, which is essentially $k^{(5/3)}$, i.e., it makes the -5/3 slope in Equation 1 appear as a flat line. Yet, all of these details are rather unimportant when compared to the 4/3 value in equation 2, which is actually written as 3/4, but constitutes the most important relationship of the paper. This is the assumption from which all of

the conclusions of this paper are drawn from. In summary, the reader should not have to consult Kaimal and Finnigan (1994) to sort out the meaning of the different ratios in this paper.

I would recommend that more quantitative techniques be used to ensure that spectra included in the analyses conform to the theoretical -5/3 slope. On suggestion is that rather than only fitting a 0th order polynomial, a better method is to fit a 1st order polynomial with statistical software, then to test for the statistical significance of the 1st order coefficient. If, for example, the p-value for the 1st order (i.e., slope) coefficient is > 0.05, it could be considered non-significant, and then it could essentially be concluded that the slope of the inertial subrange cannot be distinguished from -5/3. Once this is done, using arguments similar to model selection analysis, the 1st order polynomial model can then be reduced to the 0th order model currently described in the paper. In a manner, the goal of this is similar to the "sharpened" criteria in section 5.2/Table 2. The benefit of the statistical criteria is that it provides a defensible justification as to which 10-min periods should be included in the analysis based on their inertial subrange. Another similar idea could be to use a statistical break-point or change-point analysis for each 10-min period to determine the range over which the inertial subrange slope is -5/3 (i.e., use statistics to optimize the range of the inertial subrange for each 10-min period). This seems more complicated to me, but it could also work.

A more quantitative approach than a running mean such as in Figures 4, 5, 7, and 9 or mean/standard deviation such as in Table 2 would improve the interpretation that the results are statistical different/similar to 4/3. One suggestion I have for the figures is to replace the running mean with a local (i.e., LOESS or LOWESS) regression. This is a statistical technique that provides results similar to a running mean, but it also comes with confidence intervals. Thus, a similar figure could be produced, but with the added benefit that for any wind direction it can be tested whether or not the best fit line is significantly different from 4/3. A great example is figure 7 (right frame) where obviously for some wind directions neither the red or black lines are even close

to 4/3, but for other directions the red is similar to 4/3. A LOESS fit would give a quantitative metric to determine where this is significantly different from 4/3 and where it is significantly similar.

I would encourage the authors to reconsider their interpretation of the Huq et al. (2017) paper. While it is correct that those results suggest a magnitude of correction similar to Horst et al. (2015) (i.e., 3-7% as mentioned in their abstract, or ∼6%, which is the average of second column of their Table 2), one important distinction is that when Huq et al. (2017) applied the Kaimal (1979) and Wyngaard and Zhang (1985) corrections to their numerically simulated data, the improvement in relative error was rather small (i.e., 2.4-3.4% correction, as derived from their Table 2). Frank et al. (2016) presented data from seven sites around North and Central America where these corrections increased the w measurements by 4.5-6.8% (their Table 2). While it takes a bit of interpretation to compare the results from these papers, one interpretation could be that the numerically simulated turbulence in Huq et al. (2017) tends to produce corrections (either Kaimal or Wyngaard) that are less than what are typically observed in nature. Thus, while an overall correction of ∼6% is similar to that of Horst et al. (2015), the Kaimal/Wyngaard correction only accounts for ∼50% of this. From this perspective, the findings of Huq et al. (2017) are very similar to those of this paper, which is to say, the currently accepted CSAT3 corrections do improve w measurements, but perhaps only provide a portion of the correction that is ultimately required.

I believe that by addressing these major comments and the following specific comments listed below, that this paper will be appropriate for Atmospheric Measurement Techniques.

Specific comments:

Page 3, line 30: A better definition for "isotropic" should be given before ", which also means". The assumption of isotropic is critical for the theory that leads to the 4/3 ratio from which the entire paper is based. So, a clear definition is important.

[Figure]

Page 4, line 1: The statement "... the velocity power spectra follows the relation," is not self-evident to the casual reader. I would recommend clarifying that Kolmogorov determined this.

Page 4, line 5-6: Clarify "outer scale" Does "the most energy containing scales" refer to something similar to the peak of the spectra as shown in figure 1? Is there a way to describe the "Kolmogorov length scale", i.e., when energy dissipation begins?

Figure 1. Could add a -2/3 slope reference line for comparison.

Page 5, line 7: Does the component i refer to u, v, or w?

Page 5, lines 6-15: I found this "crude" description confusing. k2 and k3 should be defined. The Phi function should be defined. The sentence on line 15 is a repeat of an earlier statement. My big question is whether or not this section is necessary? I'm not sure it really matters much to the main understanding of the paper why the path averaging correction affects u different than v and w. At least, it might not be important enough to derive the theory behind it.

Page 6 line 7 versus Page 7 line 9: Be careful where U is defined as instantaneous versus U defined as an average over 10-minutes.

Table 1: In generally, is there a reason why H06 was only applied to 2 of the 3 datasets? This should be clarified. Also, it is not clear until Table 2 exactly which permutations of the different calculations were analyzed. It wasn't clear from Table 1 and throughout this section which different versions of these data sets were actually tested.

Page 8, lines 18 and 23: The terms "quality signal equal to zero" and "no warnings" are confusing. I am assuming these refer to the manufacturer's diagnostic value that comes from the CSAT3.

Page 8, lines 26-30: This is a strange introduction to the results. It is somewhat telling that the results are described as "we show examples". My intention by encouraging the authors to perform more rigorous statistical analysis (via 1st order polynomial p-values

or LOESS regression, etc.) is to make the results less about "examples" and more about rigorous objective metrics. The word "closely" at the end on line 28 implies some sort of goodness of fit test.

Page 8, last line on page/Page 9 line1: The first part should probably belong in the methods. For the second part, is this something that was observed from this study, or a more general finding that should have a citation?

Figure 3: Which lines does the "c" plot (i.e., 0th order polynomial) fit, w or v? It probably isn't u since that is much lower on the graph. On the caption, when it says "perpendicular", does this mean wind can flow in either direction, e.g., left-right as well as right-left? I assume the range +-10 deg means the average wind direction, not the range of instantaneous wind direction within the 10-min period?

Page 9, lines 10-12: This might be a vast overreach of the data to assume that because "both intervals" in figure 3 appear to fit within a specific inertial subrange, that it applies "irrespective of the wind conditions".

Page 9, second to last sentence: the use of 4/3 is somewhat misleading here. It really has nothing to do with the 4/3 in Equation 2. It is purely coincidence that the uncorrected Metek had a w-u ratio of $\sim$1, such that the improvement from uncorrected to corrected increases the value by $\sim$4/3. To emphasize that this value is not the same as the 4/3 in equation 2, I would simply state it was a 33% increase.

Figure 4. Why is the same graph of w-u red on the left and black on the right? If the running mean was replaced with a LOESS fit, then the confidence interval lines could also be added. In the caption, should clarify if this is the average "wind direction" over the 10-min period.

Table 2. The sharpening criteria should be mentioned earlier in the methods. Also, with a 1st-order polynomial/p-value criteria to include only 10-minute periods with no significant deviation from the -5/3 line, then the sharpening criteria would not be necessary.

Page 11, line 18: Does "lower absolute directions" mean "directions more in line with the boom"?

Page 11, line 22: The threshold +/- 0.003 seems arbitrary without some justification. The Fuw/sqrt(FuFw) <0.02 criteria should be explained in the methods with the definition of isotropy.

Page 12, line 8: This statement also applies to the Metek, although it is much smaller.

Page 12, line 19-20: These do not look that much different to me.

Figure 7: There are three different calculation scenarios presented on the right (no correction, H15, H15+H06). Which one of these applies to the left?

Page 14, line 19: The presentation of ∼0.5-1.5% is somewhat confusing. It might be simpler to describe this as "increase by 0.005-0.015", though by looking at the table this would be "0.008-0.018".

Page 14, line 22: Does this really mean that the sonic was physically rotated? This probably refers to rotating the u, v, and w measurements. Also, this methodology seems overly confusing, when it would be much simpler to reprocess the data with the planar-fit rotation.

Page 15, line 14-15: This assumes that the uncorrected portion of the w measurement is simply a scaling issue.

Page 15, line 20-21: I would remove this statement. It is far too oversimplified, and probably extremely unlikely.

Page 15, line 27-28: This is a very bold statement, but it may be justified.

Page 15, line 5 (near the bottom): Should clarify "from the corrected USA-1".

Page 16, line 26: I am confused about the verb tense. By saying "we propose to perform such an analysis" it reads like a recommendation for future research. That

is fine, but if so, a recommendation like this should probably be near the end of the conclusions.

Page 16, line 30: Similar to an earlier comment, the use of ~4/3 here is misleading because it does not have anything to do with the 4/3 in Equation 2. I would use "33% higher than the".

References:

Kaimal, J.C. and Finnigan, J.J. (1994) Atmospheric boundary layer flows: their structure and measurement, Oxford University Press, USA.

Huq, S., De Roo, F., Foken, T. and Mauder, M. (2017) Evaluation of probe-induced flow distortion of Campbell CSAT3 sonic anemometers by numerical simulation. Boundary-Layer Meteorology.

Horst, T.W., Semmer, S.R. and Maclean, G. (2015) Correction of a non-orthogonal, three-component sonic anemometer for flow distortion by transducer shadowing. Boundary-Layer Meteorology, 1-25.

Kaimal, J.C. (1979) Sonic anemometer measurement of atmospheric turbulence, pp. 551-565, Proceedings of the Dynamic Flow Conference 1978, Skovlunde, Denmark, Skovlunde, Denmark.

Wyngaard, J.C. and Zhang, S.-F. (1985) Transducer-shadow effects on turbulence spectra measured by sonic anemometers. Journal of Atmospheric and Oceanic Technology 2(4), 548-558.

Frank, J.M., Massman, W.J., Swiatek, E., Zimmerman, H.A. and Ewers, B.E. (2016) All sonic anemometers need to correct for transducer and structural shadowing in their velocity measurements. Journal of Atmospheric and Oceanic Technology 33, 149-167.

---

## Referee Comment (RC2) · J. Kochendorfer (Referee) · 31 Oct 2018

**General Comments**

"A method to assess the accuracy of sonic anemometer measurements" evaluates turbulence power spectra to estimate biases in sonic anemometer measurements. As energy is transformed from large eddies to the smallest eddies where it is finally dissipated, within a range of 'middle' sized eddies there energy flows from larger scales of turbulence to smaller scales. This middle range of turbulence is called the inertial subrange, and within it the flow of energy is relatively constant with turbulence scale. Because of this, turbulence within the inertial subrange follows predictable laws. In the manuscript these laws are used to evaluate turbulence measurements recorded using different types of sonic anemometers at different sites. This is done in part because a standard for the measure of turbulence is not readily and commonly available.

The manuscript is well written, with appropriate and clear figures, and is generally well composed. The topic is certainly worth investigating, as sonic anemometers are relied upon for measuring eddy covariance fluxes and turbulence, and many studies have cast the accuracy of their measurements into question. The technique proposed is somewhat novel, at least as a method of evaluating sonic anemometer measurements, and as such it may be useful. The technique suffers, by the authors' own admission, of being a relative measure, rather than an absolute one; the ideal ratio of 4/3 between the W and U spectra can be achieved when both W and U are incorrect, just as long as they are incorrect to the same degree. In addition, the method can only be applied to measurements that are recorded well above the surface, in well-developed turbulence, where the inertial subrange is clearly distinguishable. However the manuscript confronts these shortcomings directly, and demonstrates how the technique is still quite useful for evaluating the accuracy of sonic anemometer measurements.

**Specific Comments**

P. 2, l. 34 – 35. Although an ATI was briefly evaluated in Kochendorfer at al. (2012), Kochendorfer et al. (2012) derived their corrections using three identical R. M. Young anemometers, by changing the orientation of the center anemometer and assuming that the outer two anemometers were capable of accurately measuring the horizontal wind speed when the angle of attack was near-zero. This is the method referred to as "the third variant" used by Nakai and Shimoyama (2012) in the manuscript (l. 22 – 31), and was originally presented by Meyers and Heuer, (2006). Regarding the "busy" setup, turbulent statistics can be compared when all anemometers are oriented vertically to evaluate biases in the wind field (e.g. Kochendorfer et al., 2013).

P. 3, l. 2. Frank et al. (2013) was unique in that the anemometers were re-oriented to check for self-consistency between different measurement axes – their experiment was not similar to the Kochendorfer et al. (2012) experiment, which only used data with zero angle of attack.

P. 3, l. 4. Explain what is mean by "a combination of all three methods".

P. 3, l. 13. This is a semantic, but still significant issue: The manuscript presents a new method for evaluating biases in sonic anemometers, but it is misleading to call it a 'new reference'. For example, if two sonic anemometers differ in their measurements, this method may not necessarily be capable of determining which one is more accurate, as it does not include an independent measurement of the wind speed; it is possible that both anemometers could have a 4/3 slope, and yet still differ from each other. The manuscript would be stronger and more accurate if descriptions of the new method as a 'reference' (e.g. p. 3, l. 14 and l. 16) are reworded.

P. 3, l. 31. I'm confused by this: "all one-point correlations between velocity components become zero". This would imply that the momentum flux (u'w') is zero within the inertial subrange, but that doesn't sound possible. Please explain. Perhaps "become zero" should be reworded as "tend toward zero"?

P. 6, l. 22 and elsewhere. Change "Measurements are collected" to "Measurements were collected". Events that occurred in the past should be described using the past tense. See https://www.nature.com/scitable/topicpage/effective-writing-13815989 for examples and further explanation. All of the description of the work that was performed should be written in the past tense – this includes the majority of
5   Sections 3, 4, and 5.

P. 7, l. 2. How were the effects of the wind turbines on the spectra evaluated or ruled out? It might be worth including something in the manuscript describing the evaluation of the spectra or distances and wind directions.

Figure 3. It is probably clearer to denote the right and left panels using letters (a and b), rather than right and left. The same can be said for the other paired figure panels.

10  P. 9, l. 12. Replace "wind conditions" with "wind direction". And as Figure 4 shows, this statement isn't strictly true. I get the general idea, but perhaps it should be written more precisely.

P. 12, l. 14. "we limit the range to a close to noise-free wavenumber" is grammatically incorrect – the sentence should probably end with, "a close to noise-free wavenumber range", but then it becomes even more verbose. Rewrite the entire sentence improve clarity, brevity, and grammar. Here's a suggestion: "The wavenumber range
15  was limited to exclude noise apparent at higher wavenumbers ($k_1 > 1$ m$^{-1}$)."

P. 14, l. 17. Change, "only those spectra, which showed…" to, "only those spectra that showed…".

P. 14, l. 20 (odd break in the line numbers here, perhaps due to a premature page break or the conversion to pdf). Change "spectra are calculated" to the past tense, "spectra were calculated".

P. 15, l. 27 – 28. This is presumably only true when the measurements support the existence of a clearly defined
20  inertial subrange. It seems like a bit of a chicken and egg problem– if the inertial subrange isn't easily identified, is it because the measurements are compromised, or because the turbulence doesn't follow the textbook?

P. 15, l. 39 (last line of p.15 – another weird brake in the line numbers here). No criticism here, just a note to the authors: Many of us interested in this type of work are hoping that LIDAR measurements will still provide a true wind velocity reference – please keep working on them! Tom Horst told me about this approach long ago, and I'm
25  still waiting to see what comes of it…

**References**

Kochendorfer, J., Meyers, T. P., Frank, J. M., Massman, W. J., and Heuer, M. W.: Reply to the Comment by Mauder on "How Well Can We Measure the Vertical Wind Speed? Implications for Fluxes of Energy and Mass", Boundary-Layer Meteorology, 147, 337-345, 2013.
30  Meyers, T. P. and Heuer, M.: A field methodology to evaluate sonic anemometer angle of attack errors, 27th Conference on Agric For Meteorol, San Diego, California, 2006.

---

## Author Comment (AC1) · 17 Dec 2018

**Response to Review #2**

We also thank anonymous reviewer #2 for the constructive comments and suggestions to the manuscript. Here our response to the comments. The response is given within the %%%--- ---%%% symbols. In addition to the points raised by both reviewers, we found small errors in the calculation of the spectral ratio in Table 2 for all corrections and sharpened criteria for the CSAT3 at Risø, which have now been corrected.

Regards,
The authors

General Comments

In this paper, the authors present a novel methodology to evaluate the relative accuracy of u, v, and w measurements from a sonic anemometer by applying Kolmogorov theory to the relative magnitude of the u, v, and w spectra within the inertial subrange. Based on that theory, the v and w spectra should be 4/3 the magnitude of the u spectra. Using field data from Metek USA-1 and CSAT3 sonic anemometers at different towers, different field sites, without any shadowing correction, with shadowing correction, and with path averaging correction, the 4/3 relationship was tested. For the Metek, while the uncorrected anemometer was much lower than 4/3 for the w-to-u relationship, after applying a wind tunnel based calibration provided by the manufacturer this relationship becomes~4/3. For the uncorrected CSAT3, the w-to-u relationship is a little close to 4/3 than was the case for the uncorrected Metek, but after applying a shadowing correction, the ratio is still lower than 4/3, being ~1.2. Thus, the CSAT3 correction could be interpreted as providing only a partial amount of the correction required to achieve a 4/3 relationship.

%%%--- Thanks for this comment. This is a good way to summarize the main findings of our analysis ---%%%

This is a very novel idea, and has the potential to be fairly influential in the discipline. There are two items that seem very important that should be given more emphasis. First, this technique is among the few that does not require a comparison between one anemometer and another. Rather, it can be applied for any single anemometer at more-or-less any field site (I recognize that this is mentioned in the paper, but this is EXTRMEMELY important, so make sure there is no doubt the reader appreciates how powerful this statement is).

%%%--- We agree with the reviewer. We now include the following sentence in the abstract "and does not require the use of another measurement as reference" to further highlight the advantage of the method ---%%%

Second, because this methodology is entirely based on Kolmogorov theory about the 4/3 ration between w-u and v-u spectra, it should be emphasized that in general all of the results from the v-u tests conform to this theory. While the reader must evaluate results such as Table 2 to determine whether w-u ratios of ~ 1.0,~1.1,~1.2 are evidence of underestimated w measurements, it is crucial to note that the v-u ratios are almost entirely 4/3 for all cases. This gives a lot of credibility that the theory of isotropy is correct, and that the 4/3 standard is reasonable.

%%%--- As suggested by the reviewer, we now state for each of the cases analyzed that the v to u spectra ratios are always close to 4/3 irrespectively of the correction type or criteria used for filtering ---%%%

One improvement that will be necessary is to better clarify the differences between 4/3, 5/3, and 2/3

slopes in the inertial subrange. I was completely confused upon my first reading, and I had to consult my Kaimal and Finnigan (1994) book to sort this out. While it is clear from Equation 1 that the slope of the inertial subrange is -5/3, figure 1 is a -2/3 slope, which presumable is because it is a "frequency weighted" spectra where the y-axis is actually spectra multiplied by wave number. This is related to second sentence in section 5.1, where the frequency-weighted spectra is multiplied by $k^{(2/3)}$ to give a straight line. This is entirely confusing unless the reader recognizes that it is already multiplied by $k^{(3/3)}$, which is essentially $k^{(5/3)}$, i.e., it makes the -5/3 slope in Equation 1 appear as a flat line. Yet, all of these details are rather unimportant when compared to the 4/3 value in equation 2, which is actually written as 3/4, but constitutes the most important relationship of the paper. This is the assumption from which all of the conclusions of this paper are drawn from. In summary, the reader should not have to consult Kaimal and Finnigan (1994) to sort out the meaning of the different ratios in this paper.

%%%--- We understand the readers could be confused. We now include a statement in the caption of Fig. 1 so that the reader understands why the slope of this wavenumber pre-multiplied spectra is -2/3. When first presenting the computed spectra (for the USA-1 at Risø), we also add that these are flatten by multiplying by $k1^{2/3}$ in contrast to the spectra in Fig. 1 ---%%%

I would recommend that more quantitative techniques be used to ensure that spectra included in the analyses conform to the theoretical -5/3 slope. On suggestion is that rather than only fitting a 0th order polynomial, a better method is to fit a 1st order polynomial with statistical software, then to test for the statistical significance of the $1^{st}$ order coefficient. If, for example, the p-value for the 1st order (i.e., slope) coefficient is > 0.05, it could be considered non-significant, and then it could essentially be concluded that the slope of the inertial subrange cannot be distinguished from -5/3. Once this is done, using arguments similar to model selection analysis, the 1st order polynomial model can then be reduced to the 0th order model currently described in the paper. In a manner, the goal of this is similar to the "sharpened" criteria in section 5.2/Table 2. The benefit of the statistical criteria is that it provides a defensible justification as to which 10-min periods should be included in the analysis based on their inertial subrange. Another similar idea could be to use a statistical break-point or change-point analysis for each 10-min period to determine the range over which the inertial subrange slope is -5/3 (i.e., use statistics to optimize the range of the inertial subrange for each 10-min period). This seems more complicated to me, but it could also work.

%%%--- Although we understand that the selection of the wavenumber range did not appear quantitative, we have chosen not to follow this recommendation. As the reviewer points out, the goal of the recommendation is similar to our "sharpened" criterion, and we have chosen to focus on this selection, First, we now include the sharpened criteria to all sites in Table 2. Second, we now clarify at the beginning of section 5 (third paragraph) that the sharpened criteria are indeed used to filter out 10-min samples where the spectra do not follow the expected behavior within the inertial subrange. The strength of the sharpened criteria is that one might have slopes close to -5/3 outside the inertial subrange and so the uw-co-covariance test aids in determining the closeness of the selected range to conform to isotropy. As stated in the Discussion, several other selections have been tried with no difference in the result. See also our answer to the reviewer's specific comment "Page 11 line 22" regarding the choice of the thresholds for the sharpened criteria: there we show the sensitivity of the results to the thresholds in the sharpened criteria. ---%%%

A more quantitative approach than a running mean such as in Figures 4, 5, 7, and 9 or mean/standard deviation such as in Table 2 would improve the interpretation that the results are statistical different/similar to 4/3. One suggestion I have for the figures is to replace the running mean with a local (i.e., LOESS or LOWESS) regression.
This is a statistical technique that provides results similar to a running mean, but it also comes with confidence intervals. Thus, a similar figure could be produced, but with the added benefit that for any wind

direction it can be tested whether or not the best fit line is significantly different from 4/3. A great example is figure 7 (right frame) where obviously for some wind directions neither the red or black lines are even close to 4/3, but for other directions the red is similar to 4/3. A LOESS fit would give a quantitative metric to determine where this is significantly different from 4/3 and where it is significantly similar.

%%%--- We understand the suggestion of the reviewer. We have actually performed loess fits for the USA-1 at Risø and for the CSAT3 at Nørrekær Enge in the original manuscript (so it is now stated that these are the fits) and the estimation of the standard errors of such fits in Figs. 4, 7, and 9 and found that they were very small and difficult to discern when plotted besides the fit. For completeness, we now anyway provide some numbers related to such standard errors in the caption of the above mentioned figures ---%%%

%%%--- We agree with the suggestion of the reviewer. We have now implemented loess fits for all sites. In addition, we have now performed the estimation of the standard errors and confidence intervals of the fits for the values in Figs. 4, 7, and 9 and found that they were very small and difficult to discern when plotted besides the fit. For completeness, we now provide the standard errors in the caption of the above mentioned figures. When adding the 95% confidence interval, we have to assume that all the observations are independent, which is likely not the case. However, the observations are indeed significantly different from 4/3 for most wind directions; please see the example below for the CSAT3 at the Risø site (the grey lines show the error range for the 0.95 confidence interval) ---%%%

[Figure]

I would encourage the authors to reconsider their interpretation of the Huq et al. (2017) paper. While it is correct that those results suggest a magnitude of correction similar to Horst et al. (2015) (i.e., 3-7% as mentioned in their abstract, or ~6%, which is the average of second column of their Table 2), one important distinction is that when Huq et al. (2017) applied the Kaimal (1979) and Wyngaard and Zhang (1985) corrections to their numerically simulated data, the improvement in relative error was rather small (i.e.,2.4-3.4% correction, as derived from their Table 2). Frank et al. (2016) presented data from seven sites around North and Central America where these corrections increased the w measurements by 4.5-6.8% (their Table 2). While it takes a bit of interpretation to compare the results from these papers, one interpretation could be that the numerically simulated turbulence in Huq et al. (2017) tends to produce corrections (either Kaimal or Wyngaard) that are less than what are typically observed in nature. Thus, while an overall correction of ~6% is similar to that of Horst et al. (2015), the Kaimal/Wyngaard correction only accounts for ~50% of this. From this perspective, the findings of Huq et al. (2017) are very similar to

those of this paper, which is to say, the currently accepted CSAT3 corrections do improve w measurements, but perhaps only provide a portion of the correction that is ultimately required.

%%%--- In the introduction, we would not like to go beyond the interpretation given by Huq et al. (2017) when they summarize their findings, and therefore, we have left the text as it was in the original submission. In the Discussion we have changed the formulations to be more precise (the new version is included in the bottom of this answer).

The abstract of Huq et al. (2017) states that *"A comparison of the corrections for transducer shadowing proposed by both Kaimal et al. (Proc Dyn Flow Conf, 551–565, 1978) and Horst et al. (Boundary-Layer Meteorol 155:371–395, 2015) show that both methods compensate for **a larger part** of the observed error, but do not sufficiently account for the azimuth dependency*."

Further, the authors state in the last paragraph on page 23: "*For the standard deviation of the w-component, Horst et al. (2015) report a relative error of between 3 and 5%, which is **almost the same** as our error. We suspect the error from our numerical experiment is slightly larger because the turbulence intensity is not quite as large as in the field, where more intense turbulence tends to weaken flow-distortion effects.*"

We have written *"Huq et al. (2017) presented a novel approach for estimating the accuracy of the CSAT3 by using numerical simulations. The results of the study pointed to flow-distortion errors of **similar** magnitude as those in H15."* and suggest that this is an accurate representation based on the above quotes from Huq et al. (2017). As we state in the manuscript, our results can only be used to quantify the sonic anemometer error, if we make assumption on how the different velocity components are affected. Therefore, it is hard to say whether the findings by Huq et al. (2017) agree with our results

The section 6.2, where the Huq et al. (2017) paper is again cited has been reformulated to:
*"If we assume that the discrepancy to 4/3 is due to remaining uncorrected flow distortion and further, that flow distortion affects the observed frequencies equally, which is an assumption supported by the results presented in Huq et al. (2017), the imperfect ratios correspond directly to underestimation in the velocity variances. Since our results do not indicate how each velocity component is affected, it is still difficult to directly use the results presented here to correct the variances. However, some qualitative comparisons can be made. If, for example, the $u$- and $v$ velocity components are measured with no error, the observed ratios of 1.12--1.19 can only turn into 4/3 if the $w$ variance is increased by 18--26\%, which means that the $w$ component itself should increase by 8--12\%. This error range is in agreement with the results by Frank et al. (2016), but higher than the error suggested by Huq et al. (2017). If we, on the other hand, assume equal errors on all velocity components (positive for $u$ and $v$, and negative for $w$) the ideal ratio of 4/3 can be reached with a 4--6\% correction on the velocity components. These examples illustrate that our method can be a useful tool for judging whether flow distortion corrections of a particular sonic anemometer are adequate or not, but that it cannot be used directly to quantify the error."* ---%%%

I believe that by addressing these major comments and the following specific com-ments listed below, that this paper will be appropriate for Atmospheric Measurement Techniques.

Specific comments:

Page 3, line 30: A better definition for "isotropic" should be given before ", which also means". The assumption of isotropic is critical for the theory that leads to the 4/3 ratio from which the entire paper is based. So, a clear definition is important.

%%%--- As suggested by the reviewer we have added a description of what isotropy means and

reformulated the sentences regarding local isotropy ---%%%

Page 4, line 1: The statement "...the velocity power spectra follows the relation," is not self-evident to the casual reader. I would recommend clarifying that Kolmogorov determined this.

%%%--- We now refer the reader to Pope's 2000 textbook after the equation ---%%%

Page 4, line 5-6: Clarify "outer scale" Does "the most energy containing scales" refer to something similar to the peak of the spectra as shown in figure 1? Is there a way to describe the "Kolmogorov length scale", i.e., when energy dissipation begins?

%%%--- Scales are now added to Fig. 1 to clarify the scales in the spectra. We have also added text describing how \eta can be estimated and that this is much smaller than the sonic path length ---%%%

Figure 1. Could add a -2/3 slope reference line for comparison.

%%%--- Added as suggested by the reviewer --%%%

Page 5, line 7: Does the component i refer to u, v, or w?

%%%--- See our response to the next comment ---%%%

Page 5, lines 6-15: I found this "crude" description confusing. k2 and k3 should be defined. The Phi function should be defined. The sentence on line 15 is a repeat of an earlier statement. My big question is whether or not this section is necessary? I'm not sure it really matters much to the main understanding of the paper why the path averaging correction affects u different than v and w. At least, it might not be important enough to derive the theory behind it.

%%%--- We agree with the reviewer. The explanation of the different effects of path averaging on the three velocity components is not essential for the paper and so it is now removed ---%%%

Page 6 line 7 versus Page 7 line 9: Be careful where U is defined as instantaneous versus U defined as an average over 10-minutes.

%%%--- U was instantaneous and so to conform also to the definitions in Sect. 2.2.2, we changed "U" in Sect. 2.2.3 to Sh (and call it instantaneous horizontal wind), and replace V by S ---%%%

Table 1: In generally, is there a reason why H06 was only applied to 2 of the 3 datasets? This should be clarified. Also, it is not clear until Table 2 exactly which permutations of the different calculations were analyzed. It wasn't clear from Table 1 and throughout this section which different versions of these data sets were actually tested.

%%%--- We now include the possible permutations in Table 1 and in the caption of the table we have added "Due to the height of the instrument at Nørrekær Enge, we did not apply a PA correction as the error should be negligible" ---%%%

Page 8, lines 18 and 23: The terms "quality signal equal to zero" and "no warnings" are confusing. I am assuming these refer to the manufacturer's diagnostic value that comes from the CSAT3.

%%%--- We are now consistently referring to the manufacturer's quality signal ---%%%

Page 8, lines 26-30: This is a strange introduction to the results. It is somewhat telling that the results are described as "we show examples". My intention by encouraging the authors to perform more rigorous statistical analysis (via 1st order polynomial p-values or LOESS regression, etc.) is to make the results less about "examples" and more about rigorous objective metrics. The word "closely" at the end on line 28 implies some sort of goodness of fit test.

%%%--- We have now extended the paragraphs that introduce the results. The sharpened criteria are also firstly described here. See our response to the reviewer's general comment regarding this issue ---%%%

Page 8, last line on page/Page 9 line1: The first part should probably belong in the methods. For the second part, is this something that was observed from this study, or a more general finding that should have a citation?

%%%--- We have moved some of the lines mentioned by the reviewer to the beginning of Sect. 5 as suggested. As we now state in the same lines, normalization with U is in our study found to reduce the scatter in the velocity spectra ---%%%

Figure 3: Which lines does the "c" plot (i.e., 0th order polynomial) fit, w or v? It probably isn't u since that is much lower on the graph. On the caption, when it says "perpendicular", does this mean wind can flow in either direction, e.g., left-right as well as right-left? I assume the range +-10 deg means the average wind direction, not the range of instantaneous wind direction within the 10-min period?

%%%--- As we have now reformulated the beginning of Sect. 5, it should be clear that the fit is performed on the w spectra. We also added the information in the caption of the figure. Further, we have stated exactly at which relative directions we refer to when saying "parallel" and "perpendicular" in the caption. We now also include in the first paragraph of data treatments that we computed the mean wind direction for each 10-min period and a statement regarding what direction is meant hereafter ---%%%

Page 9, lines 10-12: This might be a vast overreach of the data to assume that because "both intervals" in figure 3 appear to fit within a specific inertial subrange, that it applies "irrespective of the wind conditions".

%%%--- The paragraph has been rephrased and now we include that this assumption is in fact tested using the sharpened criteria ---%%%

Page 9, second to last sentence: the use of 4/3 is somewhat misleading here. It really has nothing to do with the 4/3 in Equation 2. It is purely coincidence that the uncorrected Metek had a w-u ratio of ~1, such that the improvement from uncorrected to corrected increases the value by ~4/3. To emphasize that this value is not the same as the 4/3 in equation 2, I would simply state it was a 33% increase.

%%%--- We have changed all instances where 4/3 is mentioned in relation to the 3D corrected to uncorrected variances and used the suggestion by the reviewer ---%%%

Figure 4. Why is the same graph of w-u red on the left and black on the right? If the running mean was replaced with a LOESS fit, then the confidence interval lines could also be added. In the caption, should clarify if this is the average "wind direction" over the 10-min period.

%%%--- It was not the same graph but we also showed in both frames the 3D corrected w- to u-velocity spectra ratios for a better understanding of the results. However, we now show in right frame the v- to u-velocity spectra ratio for the non- and 3D-corrected data ---%%%

Table 2. The sharpening criteria should be mentioned earlier in the methods. Also, with a 1st-order polynomial/p-value criteria to include only 10-minute periods with no significant deviation from the -5/3 line, then the sharpening criteria would not be necessary.

%%%--- See our response to the general comment regarding the sharpened criteria ---%%%

Page 11, line 18: Does "lower absolute directions" mean "directions more in line with the boom"?

%%%--- We meant low relative directions. We have modified the wording according to the suggestion by the reviewer ---%%%

Page 11, line 22: The threshold +/- 0.003 seems arbitrary without some justification. The Fuw/sqrt(FuFw) <0.02 criteria should be explained in the methods with the definition of isotropy.

%%%--- The Fuw/sqrt(FuFw) criterion, i.e., the uw co-covariance is now moved to the beginning of the results section. As we responded to an earlier comment, the text describing isotropy has been extended and reformulated.

Regarding the choice of thresholds: we have now added the following text to Sect. 6.1 about the uncertainties:
"The choice of thresholds for the sharpened criteria compromised the amount of data left for the analysis; about 4\%, 25%, and 1\% of the original amount of 10-min periods for the USA-1 at Risø, CSAT3 at Risø and the CSAT3 at Nørrekær Enge, respectively. The choice, however, did not change the velocity spectra ratios significantly. The softening of the values to, e.g., 0.03 and 0.2 for the $w$-spectral slope and the $uw$-co-covariance, respectively, resulted in a change of the $w$- to $u$-velocity spectra ratio of $\approx$0.6\% for the USA-1, $\approx$1.5% for the CSAT3 at Risø, and $\approx$0.8\% for the CSAT3 at Nørrekær Enge, only"

For the reviewer's interest we have made some graphs showing the sensitivity of the w to u spectra ratios to both thresholds and the results are shown in the next three figures (for the USA-1 at Risø, the CSAT3 at Risø and the CSAT3 at Nørrekær Enge, respectively). The top frame in each plot shows the ratio as a function of the uw co-covariance (x-axis) and two thresholds for the w slope in the inertial subrange. The bottom frame is similar but showing the amount of measurements as function of the thresholds. As illustrated, the ratios at all three sites vary very little when changing these thresholds but the choice was made so that there was still enough data to be analyzed in the case where most observations were filtered out (i.e., Nørrekær Enge).

[Figure]

[Figure]

[Figure]

---%%%

Page 12, line 8: This statement also applies to the Metek, although it is much smaller.

%%%--- We have now a statement regarding this in the USA-1 analysis ---%%%

Page 12, line 19-20: These do not look that much different to me.

%%%--- They might not look that much different but for the CSAT3, the w- to u-spectra at +-180 deg is about 0.3 whereas it is less than 0.2 at Nørrekær Enge ---%%%

Figure 7: There are three different calculation scenarios presented on the right (no correction, H15, H15+H06). Which one of these applies to the left?

%%%--- As noticed by the reviewer, we now include this information in the caption of the figure ---%%%

Page 14, line 19: The presentation of ~ 0.5-1.5% is somewhat confusing. It might be simpler to describe this as "increase by 0.005-0.015", though by looking at the table this would be "0.008-0.018".

%%%--- We agree with the reviewer and have rewritten the sentence so that it reads ", which increased the CSAT3 ratios at Risø by 0.6—1.6% only ---%%%

Page 14, line 22: Does this really mean that the sonic was physically rotated? This probably refers to rotating the u, v, and w measurements. Also, this methodology seems overly confusing, when it would be much simpler to reprocess the data with the planar-fit rotation.

%%%--- We understand that a reader could think the anemometer was physically rotated, which was not.

We have therefore changed the sentence to "This was done by rotating the sonic anemometer measurements of the velocity components and applying an isotropic inertial subrange 3D spectral velocity tensor, as in H06, to calculate the nominal component spectra for this configuration." Our reluctance to use the planar-fit correction stem from earlier results published in Dellwik et al. (2010) ----%%%

Page 15, line 14-15: This assumes that the uncorrected portion of the w measurement is simply a scaling issue.

%%%--- We have reformulated the whole paragraph as we explained in the response to the major comment by the reviewer with regards to our interpretation of the work of Huq et al. (2017) ---%%%

Page 15, line 20-21: I would remove this statement. It is far too oversimplified, and probably extremely unlikely.

%%%--- We find it more unlikely that the error is only on w, and would therefore like to keep the statement in a reformulated version (see our previous response) ---%%%

Page 15, line 27-28: This is a very bold statement, but it may be justified.

%%%--- We have added ", provided that an inertial subrange is clearly apparent" to the sentence ---%%%

Page 15, line 5 (near the bottom): Should clarify "from the corrected USA-1".

%%%--- Corrected as suggested ---%%%

Page 16, line 26: I am confused about the verb tense. By saying "we propose to perform such an analysis" it reads like a recommendation for future research. That is fine, but if so, a recommendation like this should probably be near the end of the conclusions.

%%%--- Corrected as suggested ---%%%

Page 16, line 30: Similar to an earlier comment, the use of ~4/3 here is misleading because it does not have anything to do with the 4/3 in Equation 2. I would use "33% higher than the".

%%%--- Corrected as suggested ---%%%

References:

Dellwik, E., Mann, J., and Larsen, K. S.: Flow tilt angles near forest edges – Part 1: Sonic anemometry, Biogeosciences, 7, 1745-1757, https://doi.org/10.5194/bg-7-1745-2010, 2010.

---

## Author Comment (AC2) · 17 Dec 2018

**Response to Review #1**

We thank Dr. Kochendorfer for his constructive and insightful comments. Our response to each of the comments is stated. The response is given within the %%%--- ---%%% symbols below. In addition to the points raised by both reviewers, we found small errors in the calculation of the spectral ratio in Table 2 for all corrections and sharpened criteria for the CSAT3 at Risø, which have now been corrected.

Regards,
The authors

General Comments

"A method to assess the accuracy of sonic anemometer measurements" evaluates turbulence power spectra to estimate biases in sonic anemometer measurements. As energy is transformed from large eddies to the smallest eddies where it is finally dissipated, within a range of 'middle' sized eddies there energy flows from larger scales of turbulence to smaller scales. This middle range of turbulence is called the inertial subrange, and within it the flow of energy is relatively constant with turbulence scale. Because of this, turbulence within the inertial subrange follows predictable laws. In the manuscript these laws are used to evaluate turbulence measurements recorded using different types of sonic anemometers at different sites. This is done in part because a standard for the measure of turbulence is not readily and commonly available.

The manuscript is well written, with appropriate and clear figures, and is generally well composed. The topic is certainly worth investigating, as sonic anemometers are relied upon for measuring eddy covariance fluxes and turbulence, and many studies have cast the accuracy of their measurements into question. The technique proposed is somewhat novel, at least as a method of evaluating sonic anemometer measurements, and as such it may be useful. The technique suffers, by the authors' own admission, of being a relative measure, rather than an absolute one; the ideal ratio of 4/3 between the W and U spectra can be achieved when both W and U are incorrect, just as long as they are incorrect to the same degree. In addition, the method can only be applied to measurements that are recorded well above the surface, in well-developed turbulence, where the inertial subrange is clearly distinguishable. However the manuscript confronts these shortcomings directly, and demonstrates how the technique is still quite useful for evaluating the accuracy of sonic anemometer measurements.

%%%--- We acknowledge the reviewer for his general comments, all being positive. We also agree with the reviewer in that this is a very important topic as sonic anemometer measurements are the backbone of turbulence studies. As the reviewer points out, our method does suffer of being a relative measure, as we clearly stated it, but does help identifying non-accurate measurements of velocity fluctuations --- %%%

Specific Comments

P. 2, l. 34 – 35. Although an ATI was briefly evaluated in Kochendorfer at al. (2012), Kochendorfer et al. (2012) derived their corrections using three identical R. M. Young anemometers, by changing the orientation of the center anemometer and assuming that the outer two anemometers were capable of accurately measuring the horizontal wind speed when the angle of attack was near-zero. This is the method referred to as "the third variant" used by Nakai and Shimoyama (2012) in the manuscript (l. 22 – 31), and was originally presented by Meyers and Heuer, (2006). Regarding the "busy" setup, turbulent statistics can be compared when all anemometers are oriented vertically to evaluate biases in the wind

field (e.g. Kochendorfer et al., 2013).

%%%--- Indeed, we also called it "a third variant". The sentence about the Kochendorfer et al. (2012) study was moved to the same paragraph as the Nakai and Shimoyana (2012) study, to clarify that the main focus of this study was to intercompare anemometers of the same brand. We added the Meyers and Heuer (2006) study to the reference list, but note that the study is a short abstract to a conference and we rely on the editor to judge whether this is acceptable as a reference in AMT. We also added the citation to Kaimal et al. (1990), who also used sonic anemometers of the same brand mounted at a close distance to each other, to evaluate potential systematic errors. Concerning "the" busy setup, we would like to maintain a weakened version of this statement, although we agree with Dr. Kochendorfer that the observations can be compared. We merely point out that the extra booms and clamps needed for multiple sonic observations at close distance may introduce significant small-scale gradients. When looking for very small errors, it is hard to judge a priori whether such gradients can bias the result, and we see this issue as one of the very few potential problems to the investigations presented by both Kochendorfer et al. (2012) and Nakai and Shimoyama (2012). Since we have no quantitative analysis to back up this statement, we have rewritten the sentence to: "Also, it is hard to evaluate whether the somewhat ``busy" setup with several sonic anemometers in a small area could lead to additional and larger flow distortions than those using a single sonic anemometer." ---%%%

P. 3, l. 2. Frank et al. (2013) was unique in that the anemometers were re-oriented to check for self-consistency between different measurement axes – their experiment was not similar to the Kochendorfer et al. (2012) experiment, which only used data with zero angle of attack.

%%%--- We did not state that the experiment was similar, but that the *setup* was similar. In the new version of the manuscript, the citation to the Kochendorfer et al. (2012) study is moved to the paragraph where we introduce the Nakai and Shimoyama (2012) study ---%%%

P. 3, l. 4. Explain what is mean by "a combination of all three methods".

%%%--- We categorize previous work in characterizing sonic anemometer errors in three broad categories: (i) wind-tunnel calibration, (ii) comparison of different brands of sonic anemometers to each other and (iii) tilting sonic anemometers of the same brand relative to each other. "A combination of all three methods" simply means that all these methods were used in Horst et al. (2015). Since this statement is now closer to the top of the paragraph, where the three methods are stated, we hope that our wording can now be understood ---%%%

P. 3, l. 13. This is a semantic, but still significant issue: The manuscript presents a new method for evaluating biases in sonic anemometers, but it is misleading to call it a 'new reference'. For example, if two sonic anemometers differ in their measurements, this method may not necessarily be capable of determining which one is more accurate, as it does not include an independent measurement of the wind speed; it is possible that both anemometers could have a 4/3 slope, and yet still differ from each other. The manuscript would be stronger and more accurate if descriptions of the new method as a 'reference' (e.g. p. 3, l. 14 and l. 16) are reworded.

%%%--- We agree with the reviewer and have changed all "reference" entries to "method" in the revised manuscript ---%%%

P. 3, l. 31. I'm confused by this: "all one-point correlations between velocity components become zero". This would imply that the momentum flux (u'w') is zero within the inertial subrange, but that doesn't

sound possible. Please explain. Perhaps "become zero" should be reworded as "tend toward zero"?

%%%--- We agree with the reviewer that this was unclearly formulated. In the revised version it is now stated "Turbulence is locally isotropic within this range, which means that all beyond that wavenumber all one-point cross-spectra between different velocity components approach zero. For example, the cross-spectrum between u and w decreases like k1^-7/3, which is more rapid than Fu and the bulk of the momentum flux uw is located at a wavenumber lower than the inertial subrange" ---%%%

P. 6, l. 22 and elsewhere. Change "Measurements are collected" to "Measurements were collected". Events that occurred in the past should be described using the past tense. See https://www.nature.com/scitable/topicpage/effective-writing-13815989 for examples and further explanation. All of the description of the work that was performed should be written in the past tense – this includes the majority of Sections 3, 4, and 5.

%%%--- Changed as suggested by the reviewer ---%%%

P. 7, l. 2. How were the effects of the wind turbines on the spectra evaluated or ruled out? It might be worth including something in the manuscript describing the evaluation of the spectra or distances and wind directions.

%%%--- As recommended by the reviewer, in Sect. 3 we have added information regarding the wind direction sectors where possible wind turbine wakes can be found. For the CSAT3 at the Risø site, we have studied the potential effect of wakes on the observation in quite some detail and judged that the influence is negligible. First, the observation height is low compared to the hub height of the turbine and the distance to the closest turbine quite long. For the other sites, the wake sectors are harder to exclude. We now show in Figs. 4, 5, and 9 the wake sectors for the USA-1 at Risø and CSAT3 at Nørrekær Enge. The ratios computed in Table 2 now exclude all the possible wake-affected sectors for the USA-1 at Risø and for the CSAT3 at Nørrekær Enge ---%%%

Figure 3. It is probably clearer to denote the right and left panels using letters (a and b), rather than right and left. The same can be said for the other paired figure panels.

%%%--- Changed as suggested by the reviewer ---%%%

P. 9, l. 12. Replace "wind conditions" with "wind direction". And as Figure 4 shows, this statement isn't strictly true. I get the general idea, but perhaps it should be written more precisely.

%%%--- We have reformulated the sentence as recommended by the reviewer. However we did mean wind conditions because the turbulence conditions change noticeable with wind direction at this site. We have moved the sentence following that pointed out by the reviewer so that the reader understands what we mean by wind conditions ---%%%

P. 12, l. 14. "we limit the range to a close to noise-free wavenumber" is grammatically incorrect – the sentence should probably end with, "a close to noise-free wavenumber range", but then it becomes even more verbose. Rewrite the entire sentence improve clarity, brevity, and grammar. Here's a suggestion:

"The wavenumber range was limited to exclude noise apparent at higher wavenumbers ($k_1 > 1$ m$^{-1}$)."

%%%--- Corrected as suggested by the reviewer ---%%%

P. 14, l. 17. Change, "only those spectra, which showed…" to, "only those spectra that showed…".

%%%--- Corrected as suggested by the reviewer ---%%%

P. 14, l. 20 (odd break in the line numbers here, perhaps due to a premature page break or the conversion to pdf). Change "spectra are calculated" to the past tense, "spectra were calculated".

%%%--- The odd line numbers are a result of the latex style of the journal. The tense was changed as suggested by the reviewer ---%%%

P. 15, l. 27 – 28. This is presumably only true when the measurements support the existence of a clearly defined
inertial subrange. It seems like a bit of a chicken and egg problem– if the inertial subrange isn't easily identified, is it because the measurements are compromised, or because the turbulence doesn't follow the textbook?

%%%--- We agree with the reviewer that this might seem like a chicken and egg problem. In our experience, it is easy to see a well-defined inertial subrange in the velocity spectra, when this does exist. We have therefore added ", provided that an inertial subrange is clearly apparent." to the sentence --- %%%

P. 15, l. 39 (last line of p.15 – another weird brake in the line numbers here). No criticism here, just a note to the authors: Many of us interested in this type of work are hoping that LIDAR measurements will still provide a true wind velocity reference – please keep working on them! Tom Horst told me about this approach long ago, and I'm still waiting to see what comes of it…

%%%--- Thanks for the comment. We plan to submit manuscripts where we show the benefits of laser anemometry for turbulence measurements and their potential to serve as a true reference ---%%%

References

Kochendorfer, J., Meyers, T. P., Frank, J. M., Massman, W. J., and Heuer, M. W.: Reply to the Comment by Mauder on "How Well Can We Measure the Vertical Wind Speed? Implications for Fluxes of Energy and Mass", Boundary- Layer Meteorology, 147, 337-345, 2013.

Meyers, T. P. and Heuer, M.: A field methodology to evaluate sonic anemometer angle of attack errors, 27th Conference on Agric For Meteorol, San Diego, California, 2006.